# NBDI: A Simple and Effective Termination Condition for Skill Extraction from Task-Agnostic Demonstrations

**Myunsoo Kim** [* 1]  **Hayeong Lee** [* 1]  **Seong-Woong Shim** [1]  **JunHo Seo** [1]  **Byung-Jun Lee** [1 2]

## Abstract

Intelligent agents are able to make decisions based on different levels of granularity and duration. Recent advances in skill learning enabled the agent to solve complex, long-horizon tasks by effectively guiding the agent in choosing appropriate skills. However, the practice of using fixed-length skills can easily result in skipping valuable decision points, which ultimately limits the potential for further exploration and faster policy learning. In this work, we propose to learn a simple and effective termination condition that identifies decision points through a state-action novelty module that leverages agent experience data. Our approach, Novelty-based Decision Point Identification (NBDI), outperforms previous baselines in complex, long-horizon tasks, and remains effective even in the presence of significant variations in the environment configurations of downstream tasks, highlighting the importance of decision point identification in skill learning.

## 1. Introduction

The ability to make decisions based on different levels of granularity and duration is one of the key attributes of intelligence. In reinforcement learning (RL), temporal abstraction refers to the concept of an agent reasoning over a long horizon, planning, and taking high-level actions. Each high-level action corresponds to a sequence of primitive actions, or low-level actions. For example, in order to accomplish a task with a robot arm, it would be easier to utilize high-level actions such as grasping and lifting, instead of controlling every single joint of a robot arm. Temporal abstraction simplifies complex tasks by reducing the number of decisions the agent has to make, thereby alleviating the challenges that RL faces in long-horizon, sparse reward tasks.

Due to the advantages of temporal abstraction, there has been active research on developing hierarchical RL algorithms, which structure the agent's policy into a hierarchy of two policies: a high-level policy and a low level policy. The option framework (Sutton, 1998) was proposed to achieve temporal abstraction by learning options, which are high-level actions that contain inner low level policy, initiation set and termination conditions. Termination conditions are used to figure out when to switch from one option to another, enabling the agent to flexibly respond to changes in environment or task requirements. While the option framework can achieve temporal abstraction without any loss of performance when the options are optimally learned, it is usually computationally challenging to optimize for the ideal set of options within complex domains.

In this case, the skill discovery framework, which aims to discover meaningful skills (fixed-length executions of low-level policy) from the dataset through unsupervised learning techniques, has been used as an alternative. Recently, notable progress has been made in skill-based deep RL models, showing promising results in complex environments and robot manipulations (Pertsch et al., 2021a; Hakhamaneshi et al., 2021; Park et al., 2023). However, the use of fixed-length skills and the absence of appropriate termination conditions often restrict them from making decisions at critical decision points (e.g., crossroads), which can result in significant loss in performance. While there have been some studies incorporating the option framework into deep RL as is, the algorithmic complexity and unstable performance in large environments limit its widespread adoption (Kulkarni et al., 2016; Hutsebaut-Buysse et al., 2022).

In this paper, we present NBDI (Novelty-based Decision Point Identification)[1], a simple state-action novelty-based decision point identification method that allows the agent to learn terminated skills from task-agnostic demonstrations. In this context, the term task-agnostic refers to the collection of trajectories from a diverse set of tasks, excluding the

---

[*]Equal contribution  [1]Department of Artificial Intelligence, Korea University, Seoul, Republic of Korea [2]Gauss Labs Inc., Seoul, Republic of Korea. Correspondence to: Byung-Jun Lee <byungjunlee@korea.ac.kr>.

*Proceedings of the $42^{nd}$ International Conference on Machine Learning*, Vancouver, Canada. PMLR 267, 2025. Copyright 2025 by the author(s).

---

[1]Code: https://github.com/ku-dmlab/NBDI

one we are specifically interested in (see Appendix E for visualizations). Identifying critical decision points promote knowledge transfer between different tasks and stimulate exploration by closely connecting different areas in the state space (McGovern & Barto, 2001; Menache et al., 2002; Şimşek & Barto, 2004). For example, detecting doorways between rooms is useful regardless of the specific task at hand. We demonstrate the straightforward applicability of our method to the skill-based deep RL framework, and illustrate how it can lead to improvements in decision-making. To summarize, our three main contributions are as follows: (i) For the first time, we propose a skill termination condition for task-agnostic demonstrations that remains effective even when the environment configuration of complex, long-horizon downstream tasks undergo significant changes. (ii) We present a novel method in reinforcement learning, which is the identification of state-action novelty-based critical decision points. Furthermore, we demonstrate that executing terminated skills, based on state-action novelty, can substantially enhance policy learning in both robot manipulation and navigation tasks. (iii) We conduct extensive experiments, especially with other possible termination conditions, to provide insights for future research in the field of skill termination learning.

## 2. Related Works

**Option Framework** One major approach of discovering good options is to focus on identifying good terminal states, or sub-goal states. For example, landmark states (Kaelbling, 1993), reinforcement learning signals (Digney, 1998), graph partitioning (Menache et al., 2002; Şimşek et al., 2005; Machado et al., 2017a;b), and state clustering (Srinivas et al., 2016) have been used to identify meaningful sub-goal states. (Digney, 1998; Simsek et al., 2005) and (Kulkarni et al., 2016) focused on detecting bottleneck states, which are states that appear frequently within successful trajectories, but are less common in unsuccessful trajectories (e.g., a state with access door). (Şimşek & Barto, 2004) tried to identify access states, which are similar to bottleneck states, but determined based on the relative novelty score of predecessor states and successor states. Access states are found based on the intuition that sub-goals will exhibit a relative novelty score distribution with scores that are frequently higher than those of non sub-goals. These studies motivated us to search for states with meaningful properties to terminate skills. However, these methods frequently face challenges in scaling to large or continuous state spaces.

**Skill-based deep RL** As extending the classic option framework to high-dimensional state spaces through the adoption of function approximation is not straightforward, a number of practitioners have proposed acquiring skills, which are fixed-length executions of low-level policies,

to achieve temporal abstraction. For example, skill discovery (Gregor et al., 2016; Achiam et al., 2018; Mavor-Parker et al., 2022; Park et al., 2023) and skill extraction (Yang et al., 2021; Singh et al., 2020; Pertsch et al., 2021b; Hakhamaneshi et al., 2021) frameworks have proven to be successful in acquiring meaningful sets of skills. Especially, Pertsch et al. (2021a) showed promising results in complex, long-horizon tasks with sparse rewards by extracting skills with data-driven behavior priors. The learned prior enables the agent to explore the environment in a more structured manner, which leads to better performance in downstream tasks. However, we believe that their performances are greatly constrained by the use of fixed-length skills, which restricts them from making decisions at critical decision points. There are prior works introducing variable-length skill extraction methods. Salter et al. (2022) focus on learning an option-level transition model leveraging predictability to compress offline behaviors into options that terminate at bottleneck states. However, due to its model-based design, the extracted skills are not suitable for transferring to downstream tasks with significantly different environment configuration. Jiang et al. (2022) is an option framework based model that learns both options and termination condition from the task-agnostic demonstrations in terms of minimum description length (Rissanen, 1978). As it serves as a suitable benchmark for evaluating our approach, we later compare our method to Jiang et al. (2022) in Section 6.4.

**Novelty-based RL** Novelty has been utilized in reinforcement learning for various purposes. Depending on its design, novelty can be used for curiosity-driven exploration (Burda et al., 2018; Pathak et al., 2019; Sekar et al., 2020), or data coverage maximization (Bellemare et al., 2016; Hazan et al., 2019; Seo et al., 2021). It has been also used to identify sub-goals in discrete environments (Goel, 2003; Şimşek & Barto, 2004). However, to the best of our knowledge, there has been no research that has utilized state-action novelty for identifying decision points in the context of deep RL.

## 3. Background

**Markov Decision Process (MDP)** MDP is a mathematical framework to model decision making problems with discrete-time control processes. It is defined by a tuple $\{\mathcal{S}, \mathcal{A}, P, R, \gamma\}$, where $\mathcal{S}$ denotes a state space, $\mathcal{A}$ denotes a set of actions the agent can execute, $P(s'|s, a)$ denotes a transition probability, $R(s, a)$ is a reward function and $\gamma$ is a discount factor. In a MDP, the probability of transitioning to a future state depends solely on the current state, which is known as the Markov property. Given a MDP, we aim to find an optimal policy $\pi^*$ that maximizes the expected discounted sum of reward $\mathbb{E}_\pi \left[ \sum_{t=0}^{\infty} \gamma^t R(s, a) \right]$. The state value function $V^\pi(s)$ and the action value function $Q^\pi(s, a)$

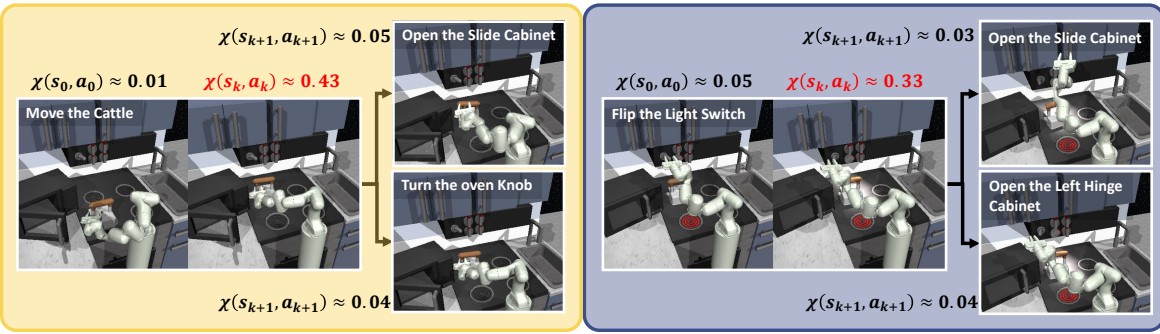

*Figure 1.* Visualization of an example of critical decision points in the kitchen environment. High state-action novelty can be found in states where a subtask has been completed, and multiple subsequent subtasks are accessible. If termination occurs at a high state-action novelty point, the agent retains multiple plausible options. For example, after moving the kettle, it may choose to open the sliding cabinet or turn the oven knob (Left). Similarly, after flipping the light switch, the agent may proceed to open either the sliding cabinet or the left-hinged cabinet (Right).

denote the conditional expectation of discounted sum of reward following policy $\pi$.

**Option Framework**  The option framework (Sutton, 1998) is one of the first studies to achieve temporal abstraction in RL. The option framework is composed of two major elements: a meta-control policy $\mu$ and a set of options $\mathcal{O}$. An option is defined as $\langle \mathcal{I}, \pi, \beta \rangle$, where $\mathcal{I} \subseteq \mathcal{S}$ defines an initiation set, $\pi : \mathcal{S} \times \mathcal{A} \to [0,1]$ defines a policy, and $\beta : \mathcal{S} \to [0,1]$ defines a termination condition. The policy $\pi$ chooses the next action, until the option is terminated by the stochastic termination condition $\beta$. Once the option terminates, the agent has an opportunity to switch to another available option at the termination state. Options usually refer to low-level polices that are promised to be good only for a subset of the state space. Thus, the presence of an appropriate initiation set $\mathcal{I}$ and termination condition $\beta$ is crucial for the agent's overall performance.

Any MDP with a fixed set of options can be classified as a Semi-Markov Decision Process (SMDP) (Sutton, 1998). SMDP (Bradtke & Duff, 1994) is an extended version of MDP for the situations where actions have different execution lengths. It serves as the foundational mathematical framework for many hierarchical RL algorithms, including the option framework.

# 4. Simple and Effective Identification of Decision Points

The option framework aims to achieve temporal abstraction by learning good options, and good options can be learned through the identification of meaningful sub-goal states (Menache et al., 2002; Şimşek & Barto, 2004), i.e., the critical decision points. In this work, we propose to use state-action novelty to identify critical decision points for skill termination, which leads to the execution of variable-length

skills. In particular, we use intrinsic curiosity module (ICM) (Pathak et al., 2017) as our state-action novelty estimator (more details in Section 6.1). However, any state-action novelty estimation mechanism that measures the joint novelty of state-action pairs can be used for our approach.

## 4.1. State-action Novelty-based Decision Point Identification

Our proposed method classifies a state-action pair with high joint state-action novelty as a decision point. A more insightful perspective on this choice can be obtained by breaking down the novelty estimator (Equation 1). By interpreting joint novelty $\chi(s, a)$ as the reciprocal of joint visitation count $N(s, a)^2$, we can decompose a state-action joint novelty $\chi$ into the product of a state novelty and a conditional action novelty. The proposed method combines the strength of both novelty estimates.

$$
\chi(s, a) = \frac{1}{N(s, a)} = \frac{1}{N(s)} \cdot \frac{1}{N(a|s)}
$$
$$
= \underbrace{\chi(s)}_{\text{state novelty}} \cdot \underbrace{\chi(a|s)}_{\text{conditional action novelty}}
$$
(1)

The state novelty $\chi(s)$ will seek for a *novel state*, which refers to a state that is either challenging to reach or rare in the dataset of agent experiences. As the skills are derived from the same pool of experiences that we use to estimate novelty, a high state novelty implies a potential lack of diverse skills to explore neighboring states effectively. Thus, increasing the frequency of decision-making in such unfamiliar states will lead to improved exploration and broader coverage of the state space when solving downstream tasks.

A conditional action novelty $\chi(a|s)$ will seek for a *novel*

---

[2]Note that although the motivation is based on pseudo counts, any novelty estimator based on $s, a$ can be used for our method.

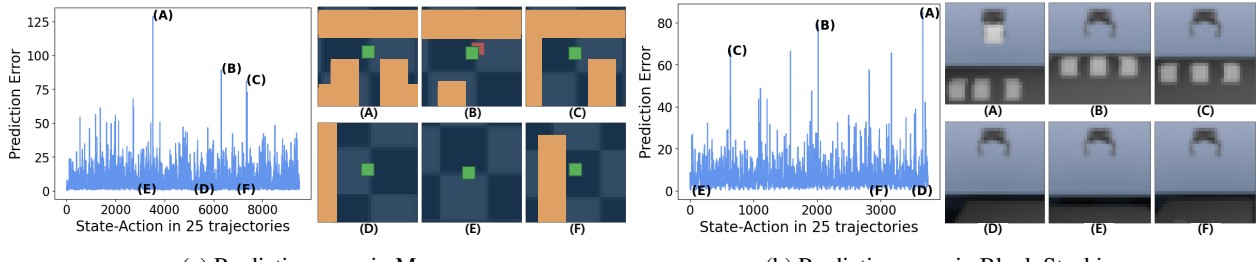

(a) Prediction error in Maze
(b) Prediction error in Block Stacking

*Figure 2.* Visualization of prediction error of ICM in maze and block stacking environment. Note the same offline data that is used to train ICM was used to compute this prediction error. (A), (B) and (C) are the state-action pairs with the highest prediction error, while (D), (E) and (F) are the ones with the lowest. Critical decision points—such as crossroads or states involving block manipulation—are typically associated with high prediction error. In contrast, low prediction errors are observed in states that are less important for decision-making.

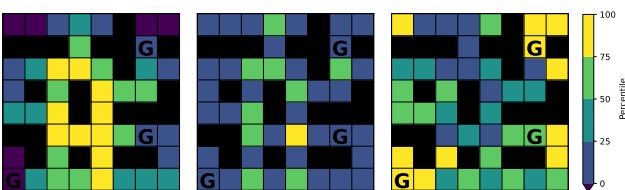

*Figure 3.* The relative frequency of termination improvement occurrences (**left**), conditional action novelty (**middle**), and state novelty (**right**) in a small grid with three different goals. Higher percentile colors indicate a relatively greater number of termination improvement occurrences, higher conditional action novelty, and higher state novelty. Further details on the visualization procedure are provided in Appendix I.1.

*action.* With the state conditioning, action novelty will be high in a state where a *multitude of actions* have been frequently executed. For example, unlike straight roads, crossroads provide the agent with options to move in multiple directions. In such states, the agent may need to perform different actions to accomplish the current goal, rather than solely depending on the current skill. This necessity arises because the current skill may have been originally designed for different goals, making it potentially less than ideal for the current goal. Guiding the agent to make more decisions in such states can increase the likelihood of solving the task at hand, ultimately accelerating the policy learning.

In the kitchen environment, as shown in Figure 1, high state-action novelty $\chi(s, a)$ tends to occur in states where a subtask has been completed. For example, after moving the kettle, the agent may choose to either open the sliding cabinet or turn the oven knob. Similarly, after completing the subtask of flipping the light switch, the agent has the option to open either the left hinge cabinet or the right slide cabinet.

In sequential manipulation tasks, such critical points are valuable because completing one subtask grants access to multiple other subtasks.

### 4.2. Termination Improvement from State-action Novelty-based Terminations

We provide an alternative interpretation on the potential benefits of identifying decision points based on state-action novelty. While maximizing skill length is advantageous in terms of temporal abstraction, extended skills can result in suboptimal behavior, especially when the skills are derived from task-agnostic trajectories. Such suboptimality of extended skills (or options) can be theoretically quantified using the termination improvement theorem (Sutton, 1998).

**Theorem 4.1.** *[Termination Improvement, (Sutton, 1998), informal] For any meta-control policy $\mu$ on set of options $\mathcal{O}$, define a new set of options $\mathcal{O}'$, which is a set of options that we can additionally choose to terminate whenever the value of a state $V^{\mu}(s)$ is larger than the value of a state given that we keep the current option $o$, $Q^{\mu}(s, o)$. With $\mu'$, which has the same option selection probability as $\mu$ but over a new set of options $\mathcal{O}'$, we have $V^{\mu'}(s) \geq V^{\mu}(s)$.*

The termination improvement theorem implies that we should terminate an option when there are much better alternatives available from the current state.

To identify the states where termination improvement occurs, we plotted the relative frequency of termination improvement occurrences in a small $8 \times 8$ grid maze with three different goal settings (Figure 3 (left)). It shows that termination improvement frequently occurs in states where diverse plausible actions exist. In states with a single available option, $V^{\mu}(s)$ would be equal to $Q^{\mu}(s, o)$. On the other hand, as more actions/options are plausible, $Q^{\mu}(s, o)$ would exhibit a broader range of values, thereby increasing the likelihood of satisfying $Q^{\mu}(s, o) < V^{\mu}(s)$. When the skills (or options) are discovered from diverse trajectories (e.g., trajectories gathered from a diverse set of goals), termination improvement is typically observed in states where a multitude of actions have been executed, such as crossroads.

However, terminating skills based on the termination improvement theorem can be challenging when the downstream task is unknown, as it requires $Q^{\mu}(s, o)$ and $V^{\mu}(s)$

**Algorithm 1** Downstream RL with NBDI

**Input:** trained low-level policy $\pi(a|z,s)$, trained novelty module $p(\beta|s,a)$, the maximum skill length $H$
**Output:** the high-level policy $\mu_\theta$

1: Initialize replay buffer $\mathcal{D}$, parameters of high-level policy $\theta$
2: $t \leftarrow 0$
3: **for** each iteration **do**
4: $\quad z_t \sim \mu_\theta(z_t|s_t)$
5: $\quad$ **for** $k = 0, 1, \ldots$ **do**
6: $\quad\quad a_{t+k} \sim \pi(a_{t+k}|z_t, s_{t+k})$
7: $\quad\quad \beta_{t+k} \sim p(\beta_{t+k}|s_{t+k}, a_{t+k})$
8: $\quad\quad$ Take action $a_{t+k}$, observe $r_{t+k}$, $s_{t+k+1}$
9: $\quad\quad$ **if** $\beta_{t+k} = 1$ or $k = H$ **then**
10: $\quad\quad\quad$ Break
11: $\quad\quad$ **end if**
12: $\quad$ **end for**
13: $\quad \tilde{r}_t \leftarrow \sum_{i=t}^{t+k} \gamma^{i-t} r_i$
14: $\quad \mathcal{D} \leftarrow \mathcal{D} \cup \{s_t, z_t, \tilde{r}_t, s_{t+k+1}, k\}$
15: $\quad$ Update $\mu_\theta(z|s)$ using RL with samples from $\mathcal{D}$
16: **end for**

to be computed in advance with the skills extracted from the downstream task trajectories. Thus, by leveraging the data collected across a diverse set of tasks, we can use conditional action novelty as a tool for pinpointing the states where a multitude of plausible actions can be taken (Figure 3 (middle)). Through experiments, we also found state novelty to be useful in terminating skills, as it encourages the agent to sufficiently explore unfamiliar parts of the state space (Figure 3 (right)). As a result, we propose to use state-action novelty, which combines the strength of both conditional action novelty and state novelty as in Equation 1, as our skill termination condition. In Section 6, we also demonstrate how these different novelty measures, utilized as termination conditions, lead to different performance outcomes.

## 5. Learning Termination Conditions through State-action Novelty Module

Our goal is to improve the learning of a new complex and long-horizon task by identifying critical decision points through a state-action novelty module. While fixed-length skills have been mostly considered for temporal abstractions in recent studies (Pertsch et al., 2021a; Hakhamaneshi et al., 2021), utilizing fixed-length skills can easily skip valuable decision points, ultimately reducing the opportunities for further exploration.

In this work, we propose to use state-action novelty as a termination condition to effectively capture critical decision points and execute terminated skills. Our approach consists

of two major steps. First, we train the state-action novelty module and then the low-level policy using task-agnostic demonstrations for skill extraction. Next, we perform online reinforcement learning with the learned variable-length skills to solve an unseen task.

**Problem Formulation** For training the state-action novelty module, we assume access to task-agnostic, expert-level demonstrations of states and actions in the form of $N$ trajectories $\mathcal{D} = \left\{ \tau^i = \{(s_t, a_t)\}_{t=0}^{T-1} \right\}_{i=0}^{N-1}$. These trajectories are collected across a diverse set of tasks except for the one we are specifically interested in (see Appendix E for more details). Since we do not make any assumptions about rewards or task labels, our model can leverage real-world datasets that can be collected at a lower cost (e.g., autonomous driving and drones).

### 5.1. Unsupervised Learning of State-action Novelty Module

In the process of unsupervised learning, our goal is to pre-train the low-level policy $\pi(a|z,s)$ and the state-action novelty module $p(\beta|s,a)$. We define a skill $z \in \mathcal{Z}$ as an embedding of state-action pairs $\tau = \{(s_i, a_i)\}_{i=t}^{t+H-1}$ and termination conditions $\boldsymbol{\beta} = \{\beta_i\}_{i=t}^{t+H-1}$. The termination conditions $\beta$ are Bernoulli random variables that decide when to stop the current skill. Through the classification of state-action pairs demonstrating significant novelty $\chi(s,a)$, $\beta$ are trained to predict the critical decision points. The point at which novelty is considered significant varies depending on the environment. In downstream tasks, the skill being executed will be terminated either when $\beta = 1$ is sampled or when the maximum skill length $H$ is reached. In all experiments, we set $H = 30$. The low-level policy is trained following standard practices (Pertsch et al., 2021a; Hakhamaneshi et al., 2021). During the training of the low-level policy, the deep latent variable model receives a randomly sampled experience from the training dataset, along with a termination condition vector provided by the state-action novelty module. The model is trained to reconstruct the corresponding action sequence and its length (i.e., the termination point) by maximizing the evidence lower bound (ELBO) (see Appendix B for details).

### 5.2. Reinforcement Learning with Skill Termination Conditions

In downstream learning, our objective is to learn a skill policy $\mu_\theta(z|s)$ that maximizes the expected sum of discounted rewards, parameterized by $\theta$. The pre-trained low-level policy $\pi(a|z,s)$ decodes a skill embedding $z$ into a series of actions, which persists until the skill is terminated by the predicted termination condition $\beta$.

The downstream learning can be formulated as a SMDP

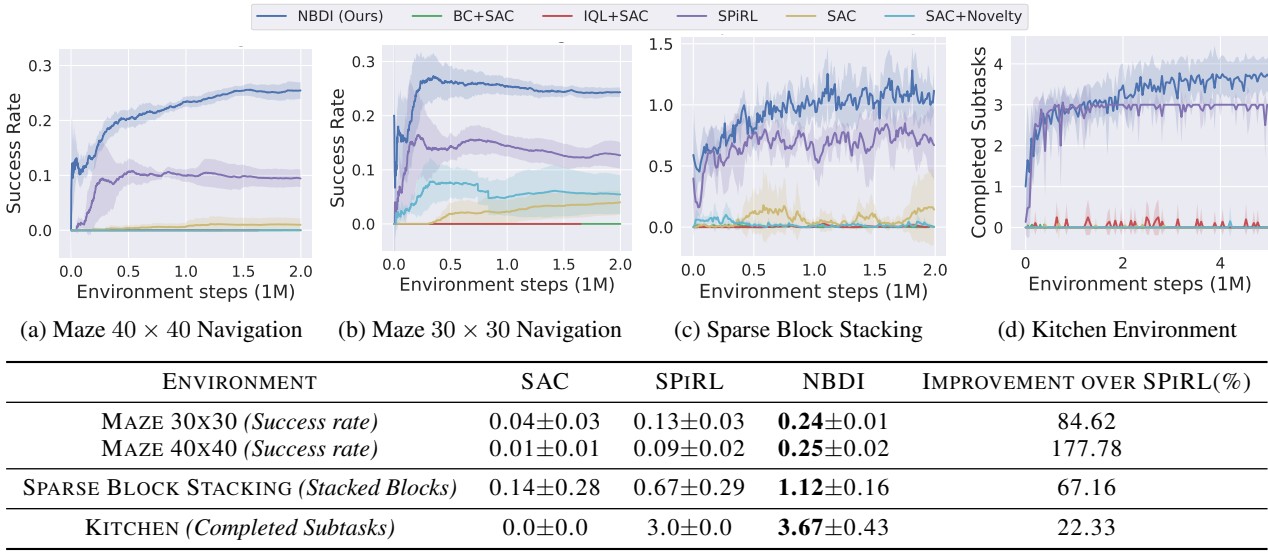

| ENVIRONMENT | SAC | SPiRL | NBDI | IMPROVEMENT OVER SPiRL(%) |
|---|---|---|---|---|
| MAZE 30X30 *(Success rate)* | 0.04±0.03 | 0.13±0.03 | **0.24**±0.01 | 84.62 |
| MAZE 40X40 *(Success rate)* | 0.01±0.01 | 0.09±0.02 | **0.25**±0.02 | 177.78 |
| SPARSE BLOCK STACKING *(Stacked Blocks)* | 0.14±0.28 | 0.67±0.29 | **1.12**±0.16 | 67.16 |
| KITCHEN *(Completed Subtasks)* | 0.0±0.0 | 3.0±0.0 | **3.67**±0.43 | 22.33 |

*Figure 5.* Performances of our method and baselines in solving downstream tasks. The shaded region represents 95% confidence interval across five different seeds. The last column of the table below illustrates the percentage improvement of our method over SPiRL.

which is an extended version of MDP that supports actions of different execution lengths. We aim to maximize discounted sum of rewards $\sum_{t \in \mathcal{T}} \tilde{r}(s_t, z_t)$ where $\mathcal{T}$ is set of time steps where we execute skills, i.e., $\mathcal{T} = \{0, k_0, k_0 + k_1, k_0 + k_1 + k_2, \ldots\}$ and $k_i$ is the variable skill length of $i$-th executed skill. The RL learning loop is described in Algorithm 1. In downstream RL tasks, we use SAC (Haarnoja et al., 2018) to update the high-level policy. More details of the learning procedure are in Appendix B.

## 6. Experiments

We design the experiments to address the following questions: (i) Does learning state-action novelty-based termination condition improve policy learning in unseen tasks? (ii) How does each component of state-action novelty contribute to the identification of critical decision points? (iii) Have we successfully identified the decision points that match our intuition? Additional studies are in Appendix A.

### 6.1. State-action Novelty Module

We utilize ICM (Pathak et al., 2017) to calculate state-action novelty for both image-based and non-image-based observations. While ICM is typically recognized for providing intrinsic motivation signals to drive exploration in online RL, we found it to be an effective state-action novelty estimator when it is pre-trained with offline trajectory datasets. Since ICM takes in state-action pair to predict next state representation, it would have high prediction error for sparse state-action pairs in the offline dataset. Figure 2 illustrates the prediction error of state-action pairs from 25 randomly selected trajectories within the offline trajectory dataset used

for training ICM. We visualized the states of the state-action pairs with high prediction error ((A), (B), (C)) and low prediction error ((D), (E), (F)) in maze environment (Figure 2a). It can be seen that high prediction error can be typically seen in states where we have multitude of plausible actions ((A), (C)) or rare state configuration (B). Note these characteristics correspond to the conditional action novelty and state novelty as illustrated in Figure 3, and leads to a high state-action novelty as in Equation 1. On the other hand, low prediction error can be seen in states where we do not have any of these properties ((D), (E), (F)). When the agent encounters a maze with an unseen goal, it would have no way of knowing which direction would lead to the goal. Therefore, encouraging the agent to make more decisions at such crossroads would effectively connect different areas within the maze, ultimately promoting exploration.

Figure 2b shows prediction error of state-action pairs in sparse block stacking environment, which is a complex robotic simulation environment that has no clearly defined subtasks. In this environment, the agent needs to stack blocks on top of each other. We can see that high prediction error occurs in states where the agent has multitude of plausible actions ((A), (B), (C)). When the robotic arm is positioned above a block, it must choose between descending to lift that block or moving towards other blocks. Likewise, when the robotic arm is holding a block, it needs to determine which block to stack it onto. This characteristic also corresponds to the conditional action novelty as depicted in Figure 3, and results in a high state-action novelty as in Equation 1. Similar to the maze environment, low prediction error occurs in states where we do not have multitude of plausible actions, or rare configuration ((D), (E), (F)).

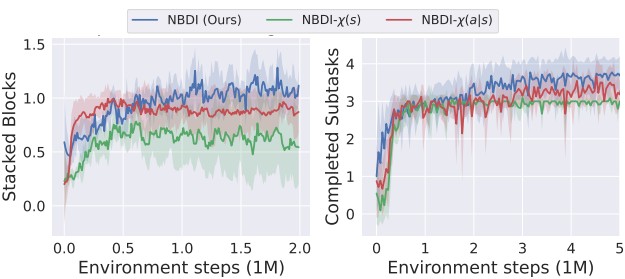
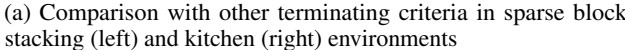
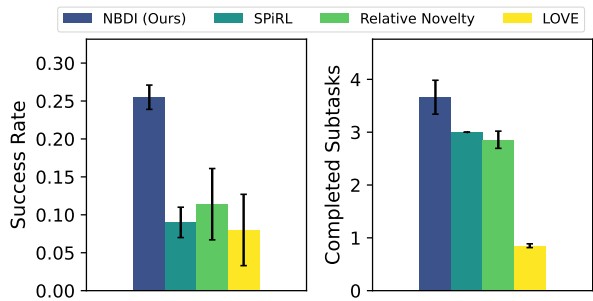

(a) Comparison with other terminating criteria in sparse block stacking (left) and kitchen (right) environments

(b) Comparison with other variable skill length methods in 40×40 maze (left) and kitchen (right) environments

*Figure 6.* Performances of our method and baselines in solving downstream tasks. The shaded region and error bar represents 95% confidence interval across five different seeds.

Thus, our findings validate that ICM can serve as an effective state-action novelty estimator when pre-trained with offline trajectory datasets.

### 6.2. Environments

Two navigation tasks (Mazes sized $30 \times 30$ and $40 \times 40$) and two simulated robot manipulation tasks (Kitchen, Sparse block stacking) are used to evaluate the performance of NBDI. All of these environments are challenging sparse reward tasks, with continuous state space and continuous action space.

A large set of task-agnostic agent experiences is collected from each environment to pre-train the low-level policy and the state-action novelty module. For downstream tasks with significantly different environment configurations (maze, sparse block stacking), we used agent-centered cropped images to extract consistent structural information from task-agnostic demonstrations. We introduce the train/transfer domain similarity for the environments used in our experiments in Appendix E. To demonstrate the effectiveness of our method on complex downstream tasks involving significant configuration changes, we evaluate the models on a maze environment with an unseen layout and a large-scale block stacking environment with a greater number of blocks in randomized positions. Both settings differ significantly from the task-agnostic offline datasets. Further details of the environments and the task-agnostic data collection procedure are in Appendix H.

### 6.3. Results

We use the following models for comparison: **Flat RL (SAC)**: Soft Actor-Critic (Haarnoja et al., 2018) agent that does not leverage prior experience. This comparison illustrates the effectiveness of temporal abstraction. **SAC+Novelty**: Flat RL that uses state-action novelty as intrinsic rewards. This comparison demonstrates the importance of incorporating state-action novelty in skill learn-

ing. **Flat Offline Learning w/ Finetuning (BC+SAC, IQL+SAC)**: The supervised behavioral cloning (BC) policy and Implicit Q-Learning (IQL) (Kostrikov et al., 2021) policy that are trained on offline data and subsequently fine-tuned for the downstream task using SAC. **Fixed-length Skill Policy (SPiRL)**: The agent that learns a fixed-length skill policy (Pertsch et al., 2021a) by leveraging prior experience. This comparison demonstrates the benefit of critical decision points identification through state-action novelty. **NBDI (Ours)**: The agent that learns a terminated skill policy through state-action novelty $\chi(s, a)$. It learns a state-action novelty based termination distribution $p(\beta|z, s)$ to predict skill termination at current step. **State Novelty Decision Point Identification (NBDI-$\chi(s)$)**: The agent that learns a terminated skill policy through state novelty. To exclusively assess the influence of the novelty type, we distilled the state-action novelty module used in NBDI into a separate network, $\chi(s)$, which solely depends on the current state. **Conditional Action Novelty Decision Point Identification (NBDI-$\chi(a|s)$)**: The agent that learns a terminated skill policy through conditional action novelty $\frac{\chi(s,a)}{\chi(s)}$, where $\chi(s)$ is the distilled state novelty module used for NBDI-$\chi(s)$. Implementation details of the baselines are in Appendix I.

In both the robot manipulation tasks and the navigation tasks, executing terminated skills through state-action novelty (NBDI-$\chi(s, a)$) facilitates convergence toward a more effective policy (Figure 5). While NBDI manages to accomplish all four subtasks in the kitchen environment, others never achieves the maximum return. Furthermore, as shown the table in Figure 5, NBDI surpasses SPiRL even within a challenging robotic simulation environment where there are no clearly defined subtasks (Sparse block stacking). However, SAC, SAC+Novelty, BC+SAC and IQL+SAC show poor performance due to their lack of temporal abstraction, which limits their ability to explore unseen tasks effectively.

In alignment with our motivation for state-action novelty, conditional action novelty (NBDI-$\chi(a|s)$) appears to play a crucial role in identifying decision points (Figure 6a).

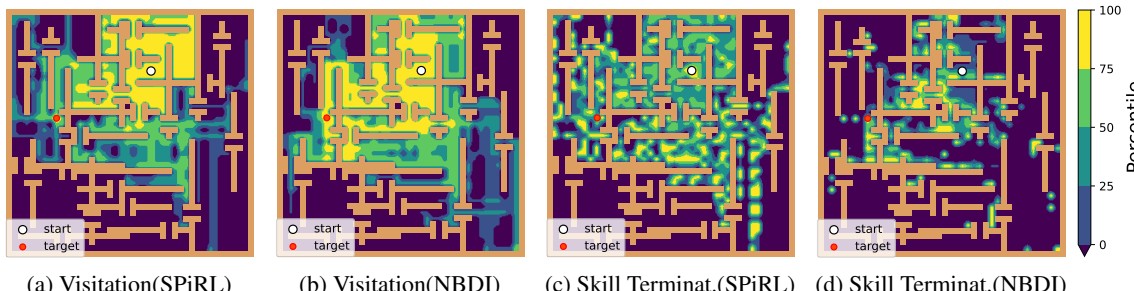

(a) Visitation(SPiRL)     (b) Visitation(NBDI)     (c) Skill Terminat.(SPiRL)     (d) Skill Terminat.(NBDI)

*Figure 7.* Visualization of decision points made by SPiRL and NBDI in the maze environment. We sampled 100 trajectories for each trained policy to observe the points at which they make decisions. Higher percentile colors suggest a relatively greater number of visitation frequencies and termination frequencies.

*Table 1.* Success rate of NBDI and SPiRL with offline trajectories generated by mediocre-level policy with weighted Gaussian random noise in maze environment.

| DATASET QUALITY | NBDI | SPiRL |
|---|---|---|
| STOCHASTIC BC ($\sigma = 0.5$) | **28**% | 0% |
| STOCHASTIC BC ($\sigma = 0.75$) | **22**% | 0% |

While it appears that terminating skills solely based on state novelty doesn't lead to better performance, combining it with conditional action novelty (resulting in state-action novelty) leads to better exploration and better convergence.

Figure 7 compares decision points made by SPiRL and NBDI in the maze environment. This result provides the answer to our third question. While the SPiRL agent makes decisions in random states, our model tends to make decisions in crossroad states or states that are unfamiliar. For instance, in the lower-right area of the maze, SPiRL shows periodic skill terminations due to its fixed-length of skills, whereas our approach tends to make decisions in states characterized by high conditional action novelty or state novelty.

### 6.4. Comparison to Other Variable-length Skill Extraction Methods

We compare the performance of NBDI, relative novelty (Şimşek & Barto, 2004) and LOVE (Jiang et al., 2022) in the maze and kitchen environment (Figure 6b). Relative novelty identifies termination condition based on the assumption that sub-goals typically show a relative novelty score distribution with higher scores than those of non-sub-goals. LOVE is an option framework based model that learns both options and termination condition from the task-agnostic demonstrations in terms of minimum description length. Since LOVE is designed for discrete action space, we used an MLP layer as the action decoder to handle continuous actions. Furthermore, since LOVE was only evaluated in downstream tasks that has the same MDP as the data collection environment (except the reward function), we evaluated whether it can transfer to downstream tasks with signifi-

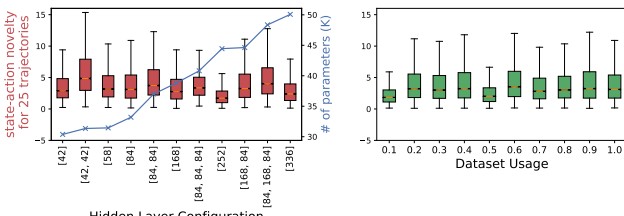

*Figure 8.* Illustration of the impact of varying the width and depth configuration of hidden layer and dataset usage on state-action novelty estimation. (Left) shows how varying hidden layer configurations (depth and width) influences the estimated state-action novelty (box plots) and the total number of parameters (blue line). The x-axis lists different hidden layer setups. (Right) illustrates how varying the proportion of dataset usage affects novelty estimation.

cantly different environment configuration (Maze $40 \times 40$). Implementation details of baselines are in Appendix I.4.

As shown in Figure 6b, LOVE performs comparably to SPiRL in the maze environment, demonstrating its applicability to continuous action spaces. However, its underperformance compared to our method suggests that the variable skills learned by LOVE do not effectively generalize to downstream tasks with significantly different environment configuration. Moreover, we observed that LOVE's performance significantly declines as the complexity of the action space increases (kitchen). NBDI's high performance compared to these baseline methods illustrates the need for a more effective termination condition for skill models in solving such challenging tasks.

### 6.5. Critical Decision Points with Suboptimal Data

We investigate how the quality of offline data affects the performance of our approach. We trained a behavior cloning (BC) policy on expert-level trajectories to generate mediocre quality demonstrations. We additionally added weighted Gaussian random noise ($\sigma$) to actions of BC policy to add stochasticity to the generated dataset. Table 1 shows that even in a less challenging goal setting compared to Figure

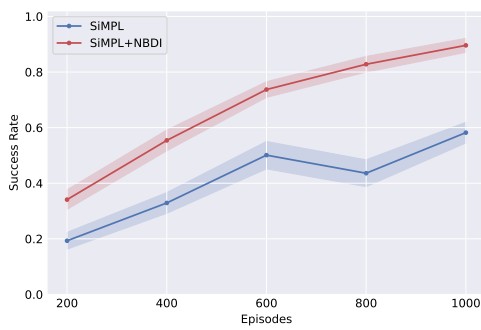

*Figure 9.* Comparison of success rates in the maze environment during the skill-based meta-training phase of SiMPL across 10 meta-training tasks (10 random goal locations). The shaded regions indicate standard errors over three random seeds.

7, SPiRL fails to reach the goal, while NBDI achieves a success rate of 28% and 22%. However, as the policy generating the trajectory becomes more stochastic (Table 1), it gathers data primarily around the initial state, leading to an overall reduction in the scale of prediction errors. Thus, we can see that the level of stochasticity in the dataset influences critical decision point detection, which remains a limitation of our work. Visualizations of decision points and prediction error with suboptimal dataset are in Appendix C.

### 6.6. Model Capacity and Critical Decision Point Detection

To assess how the model capacity and dataset utilization influence critical point detection, we varied the width and depth of the neural network across different settings. Figure 8 (left) shows that the estimated state-action novelty by ICM is not affected by the number of parameters used to train the model. Figure 8 (right) demonstrates that the scale of the estimated state-action novelty remains consistent even as the dataset size decreases. We can see that our proposed approach for detecting critical decision points is robust to various number of parameters or size of datasets.

## 7. Applying NBDI to Skill-based Meta Reinforcement Learning

To explore the broader applicability of our approach, we investigated its compatibility with a skill-based approach, SiMPL (Nam et al., 2022) that leverages task-agnostic demonstrations to solve challenging long-horizon, sparse-reward meta-RL tasks. Specifically, we apply our method during the skill extraction phase to learn variable-length skills in place of fixed-length skills. In the maze environment, we randomly sampled 10 goal locations for meta-training. The table below compares the success rate of meta-policies trained with SiMPL (fixed-length skills) and our method during the meta-training phase across episodes. In

| | ep20 | ep100 | ep300 | ep500 |
|---|---|---|---|---|
| SiMPL | 0.143±0.063 | 0.593±0.191 | 0.667±0.193 | 0.990±0.006 |
| SiMPL+NBDI | **0.560**±0.121 | **0.960**±0.021 | **0.980**±0.011 | **0.993**±0.003 |

*Table 2.* Comparison of success rates in the maze environment during the target task learning phase of SiMPL with standard errors over three random seeds.

Figure 9, the result shows that the extracted variable-length skills allows the meta-policy to better promote knowledge transfer between different tasks, helping the meta-policy in combining the skills to complete complex tasks. We report mean success rates on the 10 meta-training tasks across 3 different seeds with standard errors. Furthermore, during the target task learning phase, the meta-policy learned through our approach leads to significantly better sample efficiency on the unseen target task. Results in Table 2 indicate that our approach can be effectively integrated with a broader class of skill-based methods that leverage task-agnostic demonstrations.

## 8. Conclusion

We present NBDI, an approach for learning terminated skills through a state-action novelty module that leverages offline, task-agnostic datasets. Our approach significantly outperforms previous baselines in solving complex, long-horizon tasks and shows effectiveness even under significant changes in environment configuration of downstream tasks. A promising direction for future work is to use novelty-based decision point identification to learn variable-length skills in offline skill execution (Ajay et al., 2020; Hakhamaneshi et al., 2021).

## Acknowledgments

This work was partly supported by Institute of Information & communications Technology Planning & Evaluation (IITP) grant funded by the Korea government (MSIT) (No. RS-2022-II220311, Development of Goal-Oriented Reinforcement Learning Techniques for Contact-Rich Robotic Manipulation of Everyday Objects, No. RS-2024-00457882, AI Research Hub Project, and No. RS-2019-II190079, Artificial Intelligence Graduate School Program (Korea University)), the IITP(Institute of Information & Coummunications Technology Planning & Evaluation)-ITRC(Information Technology Research Center) grant funded by the Korea government(Ministry of Science and ICT)(IITP-2025-RS-2024-00436857), the NRF (RS-2024-00451162) funded by the Ministry of Science and ICT, Korea, BK21 Four project of the National Research Foundation of Korea, and the National Research Foundation of Korea (NRF) grant funded by the Korea government (MSIT)(RS-2025-00560367), and the IITP under the Artificial Intelligence Star Fellowship

support program to nurture the best talents (IITP-2025-RS-2025-02304828) grant funded by the Korea government (MSIT).

## Impact Statement

This paper presents work whose goal is to advance the field of Reinforcement Learning. There are many potential societal consequences of our work, none which we feel must be specifically highlighted here.

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

## A. Ablation

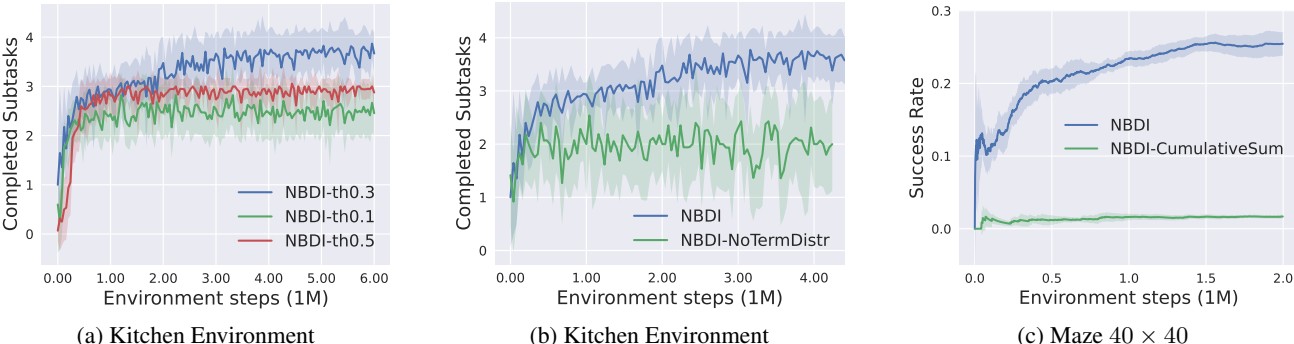

(a) Kitchen Environment    (b) Kitchen Environment    (c) Maze $40 \times 40$

*Figure 10.* Ablation in variable-length skills (a), no termination distribution (b), and criteria to determine decision points (c). The shaded region represents 95% confidence interval across five different seeds.

### A.1. Ablation in NBDI

Figure 10a compares the performance of our model (NBDI-th0.3) in the kitchen environment with different state-action novelty threshold values. We can see that there is no significant improvement in performance compared to SPiRL when the threshold value is not appropriately chosen. For example, as illustrated in Figure 11, termination distributions learned with low threshold values can disturb the policy learning by terminating skills in states that lack significance. It illustrates that threshold value needs to be appropriately chosen to capture meaningful decision points.

### A.2. Ablation in No Termination Distribution

Figure 10b shows the performance drop when we do not learn the termination distribution in advance (see Appendix B for more details). NBDI-NoTermDistr directly uses the state-action novelty module in the downstream learning phase to terminate skills. The performance gap indicates that the skill embedding space in offline skill extraction learning needs to be learned with terminated skills to effectively guide the agent in choosing variable-length skills. Thus, it is necessary to jointly optimize the termination distribution, skill embedding space, and skill prior using the deep latent variable model in offline skill extraction.

### A.3. Ablation in Criteria to Determine Decision Points

Figure 10c shows the performance difference when we use cumulative sum of state-action novelties to learn decision points. NBDI-CumulativeSum terminates skills once the cumulative sum of state-action novelty reaches or surpasses a predefined threshold. This comparison implies that accumulating novelties does not lead to the identification of significant termination points.

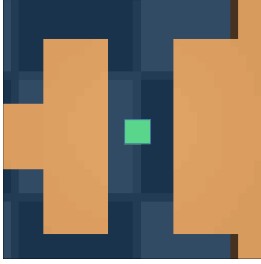

*Figure 11.* A bad example of decision point in the maze environment.

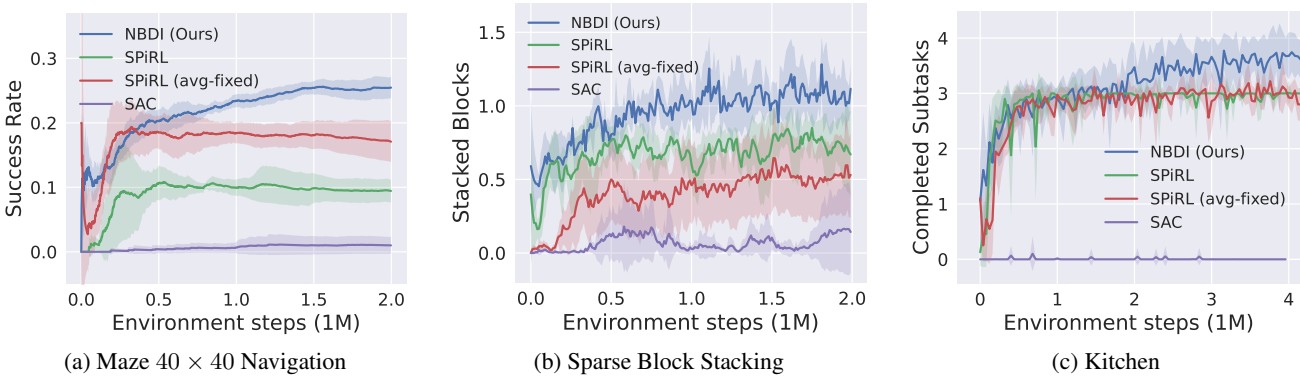

(a) Maze $40 \times 40$ Navigation    (b) Sparse Block Stacking    (c) Kitchen

*Figure 12.* Performances of our method and baselines in solving downstream tasks. The shaded region represents 95% confidence interval across five different seeds.

### A.4. Comparison to SPiRL with Fixed Average Skill Length of NBDI

Figure 12 shows whether NBDI still outperforms SPiRL when SPiRL uses average skill length of NBDI (SPiRL (avg-fixed)). The average lengths of skills of NBDI was 26 in the maze environment, 22 in the sparse block stacking environment, and 25 in the kitchen environment. We found NBDI still outperforms SPiRL with those fixed average skill lengths. We also observed that SPiRL, when set to those average skill lengths, performs worse than SPiRL configured with a fixed skill length of 10 in the block stacking environment, and better in the maze environment. However, across any fixed skill length ranging from 10 to 30, there was no instance where SPiRL outperformed NBDI. This demonstrates that our model can effectively leverage critical decision points in the environment compared to fixed length approaches.

To investigate how frequently the maximum skill length gets reached, we tracked the skill lengths of high-level policy for each environment during the downstream reinforcement learning process. The percentages of executed skill lengths shorter than $H$ are as follows: 19.8% in the maze environment, 30.8% in the sparse block stacking environment, and 17.4% in the kitchen environment. When categorizing the skill lengths into intervals of 1–10, 11–20, and 21–30, the distributions are as follows: 13.2%, 2.8%, and 84% for the maze environment; 28.6%, 1.6%, and 69.8% for the sparse block stacking; and 16.8%, 0%, and 83.2% for the kitchen.

It can be observed that, for the majority of the time, our high-level policy maximizes temporal abstraction by executing longer skills (ranging from 21 to 30). However, our method also allows the high-level policy to capture important decision points through shorter skills (ranging from 1 to 10), promoting more efficient exploration of the state space and enhancing the transfer of knowledge across various tasks.

*Table 3.* Truncated skill lengths generated by our method across various tasks with a maximum skill length of $H = 30$.

| SKILL LENGTH $h$ | MAZE | SPARSE BLOCK STACKING | KITCHEN |
|---|---|---|---|
| $h < H$ | 19.8% | 30.8% | 17.4% |
| $1 \leq h \leq 10$ | 13.2% | 28.6% | 16.8% |
| $11 \leq h \leq 20$ | 2.8% | 1.6% | 0% |
| $21 \leq h \leq H$ | 84% | 69.8% | 83.2% |

## B. Skill Extraction from Demonstration in NBDI

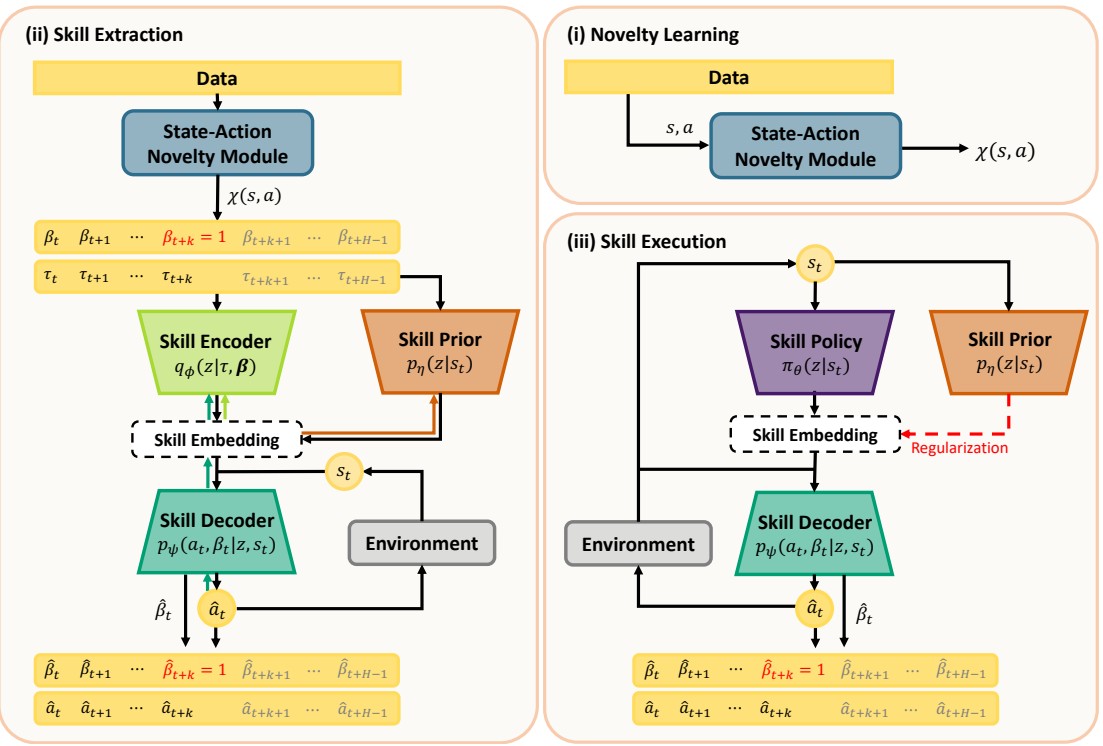

*Figure 13.* Novelty-based Decision Point Identification (NBDI), has two main procedures: (i) **novelty learning and skill extraction:** training the state-action novelty model and learning the skill prior, skill embedding space and termination distribution with the pre-trained novelty model. (ii) **skill execution**: performing reinforcement learning with termianted skills to solve an unseen task.

### B.1. Learning the Skill Prior, Skill Embedding Space and Termination Distribution

To learn the skill embedding space $\mathcal{Z}$, we train a latent variable model consisting of a Long short-term memory (LSTM) (Hochreiter & Schmidhuber, 1997) encoder $q_\phi(z|\tau, \boldsymbol{\beta})$ and a decoder $p_\psi(a_t, \beta_t|z, s_t)$. To learn model parameters $\phi$ and $\psi$, the latent variable model receives a randomly sampled experience $\tau$ from the training dataset $\mathcal{D}$ along with a termination condition vector $\boldsymbol{\beta}$ from the state-action novelty module, and tries to reconstruct the corresponding action sequence and its length (i.e., point of termination) by maximizing the evidence lower bound (ELBO):

$$\log p(a_t, \beta_t|s_t) \geq \mathbb{E}_{z \sim q_\phi(z|\tau, \boldsymbol{\beta}), \tau \sim \mathcal{D}} [\underbrace{\log p_\psi(a_t, \beta_t|z, s_t)}_{\mathcal{L}_{\text{rec}}(\phi, \psi)} + \alpha \underbrace{(\log p(z) - \log q_\phi(z|\tau, \boldsymbol{\beta})}_{\mathcal{L}_{\text{reg}}(\phi)})] \tag{2}$$

where $\alpha$ is used as the weight of the regularization term (Higgins et al., 2016). The Kullback-Leibler (KL) divergence between the unit Gaussian prior $p(z) = \mathcal{N}(0, I)$ and the posterior $\log q_\phi(z|\tau, \boldsymbol{\beta})$ makes smoother representation of skills.

To offer effective guidance in selecting skills for the current state, the skill prior $p_\eta(z|s_t)$, parameterized by $\eta$, is trained by minimizing its KL divergence from the predicted posterior $q_\phi(z|\tau, \boldsymbol{\beta})$. In the context of the option framework, it can also be viewed as the process of obtaining an appropriate initiation set $\mathcal{I}$ for options/skills. This will lead to the minimization of the prior loss:

$$\mathcal{L}_{\text{prior}}(\eta) = \mathbb{E}_{\tau \sim \mathcal{D}} [D_{KL}(q_\phi(z|\tau, \boldsymbol{\beta}) \| p_\eta(z|s_t))] \tag{3}$$

The basic architecture for skill extraction and skill prior follows prior works (Pertsch et al., 2021a; Hakhamaneshi et al., 2021), which have proven to be successful. In summary, termination distribution, skill embedding space, and skill prior are jointly optimized with the following loss:

$$\mathcal{L}_{\text{total}} = \mathcal{L}_{\text{rec}}(\phi, \psi) + \alpha \mathcal{L}_{\text{reg}}(\phi) + \mathcal{L}_{\text{prior}}(\eta) \tag{4}$$

---

**Algorithm 2** Reinforcement learning with NBDI

---

**Input:** trained skill decoder $p_\psi(a, \beta|z, s)$, discount factor $\gamma$, target divergence $\delta$, learning rates $\lambda_\pi, \lambda_Q, \lambda_\omega$, target update rate $\epsilon$

1: Initialize replay buffer $\mathcal{D}$, high-level policy $\pi_\theta(z|s)$, critic $Q_\xi(s, z)$, target network $\bar{\xi} = \xi$
2: **for** each iteration **do**
3:     **for** each environment step **do**
4:         $z_t \sim \pi_\theta(z_t|s_t)$
5:         **for** $k = 0, 1, \ldots$ **do**
6:             $a_{t+k}, \beta_{t+k} \sim p_\psi(a_{t+k}, \beta_{t+k}|z_t, s_{t+k})$
7:             $s_{t+k+1} \sim p(s_{t+k+1}|s_{t+k}, a_{t+k})$
8:             **if** $\beta_{t+k} = 1$ or $k = H$ **then**
9:                 Break
10:             **end if**
11:         **end for**
12:         $\tilde{r}_t \leftarrow \sum_{i=t}^{t+k} \gamma^{i-t} R(s_i, a_i)$
13:         $\mathcal{D} \leftarrow \mathcal{D} \cup \{s_t, z_t, \tilde{r}_t, s_{t+k+1}, k\}$
14:     **end for**
15:     **for** each gradient step **do**
16:         $z_{t+k+1} \sim \pi_\theta(z_{t+k+1}|s_{t+k+1})$
17:         $\bar{Q} = \tilde{r}_t + \gamma^k \left[ Q_{\bar{\xi}}(s_{t+k+1}, z_{t+k+1}) - \omega D_{KL}(\pi_\theta(z_{t+k+1}|s_{t+k+1}) \| p_\eta(z_{t+k+1}|s_{t+k+1})) \right]$
18:         $\theta \leftarrow \theta - \lambda_\pi \nabla_\theta \left[ Q_\xi(s_t, z_t) - \omega D_{KL}(\pi_\theta(z_t|s_t) \| p_\eta(z_t|s_t)) \right]$
19:         $\phi \leftarrow \xi - \lambda_Q \nabla_\xi \left[ \frac{1}{2}(Q_\xi(s_t, z_t) - \bar{Q})^2 \right]$
20:         $\omega \leftarrow \omega - \lambda_\omega \nabla_\omega \left[ \omega \cdot (D_{KL}(\pi_\theta(z_t|s_t) \| p_\eta(z_t|s_t)) - \delta) \right]$
21:         $\bar{\xi} \leftarrow \epsilon \xi + (1 - \epsilon)\bar{\xi}$
22:     **end for**
23: **end for**
24: **return** trained policy $\pi_\theta(z_t|s_t)$

---

*Table 4.* Skill Prior Hyperparameters

| HYPERAPARAMETER | VALUE |
|---|---|
| BATCH SIZE | 16 |
| OPTIMIZER | RADAM($\beta_1 = 0.9, \beta_2 = 0.999, lr = 1e-3$) |
| REGULARIZATION WEIGHT $\alpha$ | 1.0 |
| SKILL ENCODER | |
|    DIM-$\mathcal{Z}$ IN VAE | 32 |
|    HIDDEN DIM | 128 |
|    # LSTM LAYERS | 1 |
| SKILL PRIOR (KITCHEN) | |
|    HIDDEN DIM | 128 |
|    # FC LAYERS | 6 |
| SKILL PRIOR (MAZE, BLOCK STACKING) | |
|    KERNEL SIZE | (4, 4) |
|    CHANNELS | 8, 16, 32 |
|    # CONVOLUTION LAYERS | 3 |
| SKILL DECODER | |
|    HIDDEN DIM | 128 |
|    # HIDDEN LAYERS | 6 |

## B.2. Reinforcement Learning with NBDI

In downstream learning, our objective is to learn a skill policy $\pi_\theta(z|s_t)$ that maximizes the expected sum of discounted rewards, parameterized by $\theta$. The pre-trained decoder $p_\psi(a_t, \beta_t|z, s_t)$ decodes a skill embedding $z$ into a series of actions, which persists until the skill is terminated by the predicted termination condition $\beta_t$. The downstream learning can be formulated as a SMDP which is an extended version of MDP that supports actions of different execution lengths.

Adapted from Soft Actor-Critic (SAC) (Haarnoja et al., 2018), we aim to maximize discounted sum of rewards while minimizing its KL divergence from the pre-trained skill prior on SMDP. The regularization weighted by $\omega$ effectively reduces the size of the skill latent space the agent needs to explore.

$$J(\theta) = \mathbb{E}_\pi \left[ \sum_{t \in \mathcal{T}} \tilde{r}(s_t, z_t) - \omega D_{\mathrm{KL}}\big( \pi(z_t|s_t), p_\eta(z_t|s_t) \big) \right] \tag{5}$$

where $\mathcal{T}$ is set of time steps where we execute skills, i.e., $\mathcal{T} = \{0, k_0, k_0 + k_1, k_0 + k_1 + k_2, \ldots\}$ where $k_i$ is the variable skill length of $i$-th executed skill.

To handle actions of different execution lengths, the following Q-function objective is used:

$$J_Q(\xi) = \mathbb{E}_{(s_t, z_t, \tilde{r}_t, s_{t+k+1}, k) \sim \mathcal{D}, z_{t+k+1} \sim \pi_\theta(\cdot|s_{t+k+1})} \left[ \frac{1}{2} (Q_\xi(s_t, z_t) - \bar{Q})^2 \right],$$

$$\text{where} \quad \bar{Q} = \tilde{r}_t + \gamma^k [Q_{\bar\xi}(s_{t+k+1}, z_{t+k+1}) - \omega D_{KL}(\pi_\theta(z_{t+k+1}|s_{t+k+1}) \| p_\eta(z_{t+k+1}|s_{t+k+1}))]$$

$\omega$ represents the temperature for KL-regularization, $k$ denotes the number of time steps elapsed from the start state $s_t$ to the termination state $s_{t+k+1}$, and $\tilde{r}$ represents the cumulative discounted reward over the $k$ time steps. The detailed RL learning loop is described in Algorithm 2 and Figure 13.

## B.3. Threshold for environments

In practice, the thresholds we tuned through experiments (see Appendix A.1) approximately correspond to the 97th percentile of the novelty values computed over task-agnostic demonstrations—e.g., kitchen = 0.3 (97.47th percentile), maze = 50 (96.86th percentile), and block stacking = 40 (96.12th percentile). We hope the percentile-based guidance would help the readers more easily apply our method across new environments.

*Table 5.* Downstream RL Hyperparameters

| HYPERAPARAMETER | VALUE |
|---|---|
| **DOWNSTREAM REINFORCEMENT LEARNING** | |
| BATCH SIZE | 256 |
| OPTIMIZER | ADAM($\beta_1 = 0.9, \beta_2 = 0.999, lr = 3e-4$) |
| REPLAY BUFFER SIZE | 1E6 |
| DISCOUNT FACTOR $\gamma$ | $0.99^{\frac{1}{10}}$ |
| TARGET NETWORK UPDATE RATE $\epsilon$ | $5e-3$ |
| TARGET DIVERGENCE $\delta$ | 5 (KITCHEN), 1 (MAZE, SPARSE BLOCK STACKING), |
| **VARIABLE LENGTH SKILL** | |
| THRESHOLD OF NOVELTY | 0.3 (KITCHEN), 50 (MAZE), 40 (SPARSE BLOCK STACKING) |
| MAXIMUM SKILL LENGTH $H$ | 30 |

## C. Visualization of Critical Decision Points with Suboptimal Data

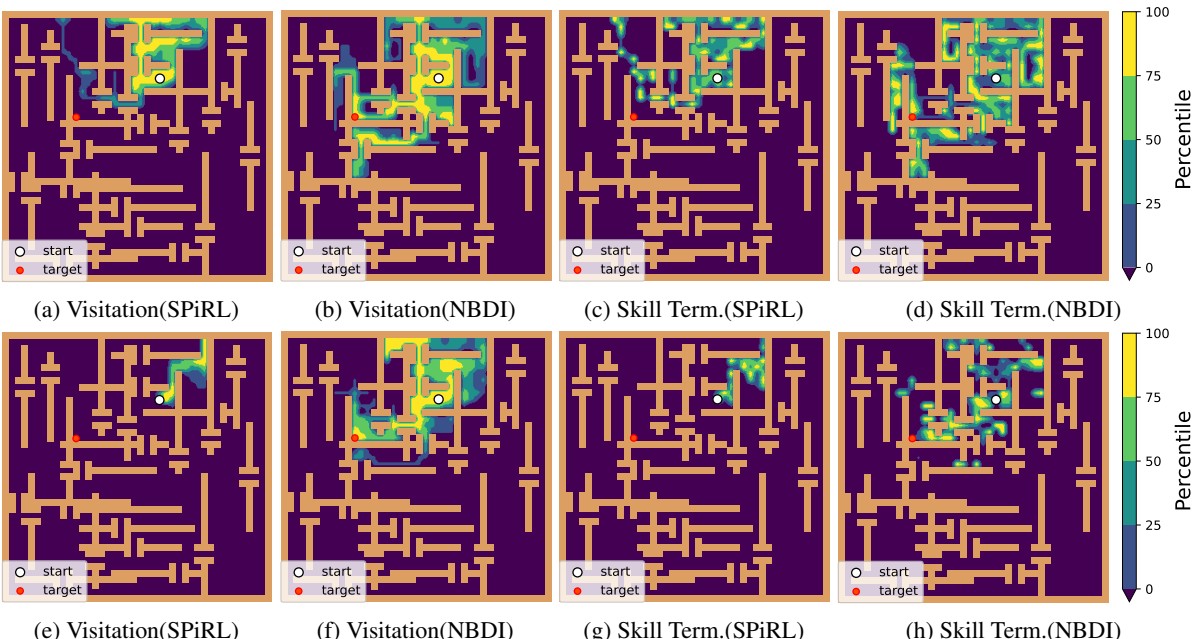

*Figure 14.* Visualization of visitations and decision points made by SPiRL and NBDI in the maze environment (**top**: trained with stochastic BC ($\sigma = 0.5$) dataset, **bottom**: trained with stochastic BC ($\sigma = 0.75$) dataset). We sampled 100 trajectories for each trained policy to observe the points at which they make decisions. Higher percentile colors suggest a relatively greater number of visitation frequencies and termination frequencies. Note that the termination frequencies are normalized by the overall visitation frequencies for better visualization.

Figure 14 and Figure 15 (top, middle) show that with suboptimal dataset, NBDI is still able to learn termination points characterized by high conditional action novelty or state novelty. Figure 14a and Figure 14e shows that SPiRL can only navigate around the initial state using fixed-length skills extracted from the suboptimal dataset, whereas NBDI can successfully reach the goal efficiently (Figure 14b and Figure 14f).

However, with dataset generated by random walk (Figure 15 (bottom)), it becomes challenging to learn meaningful decision points. As the policy generating the trajectory becomes more stochastic, it gathers data primarily around the initial state, leading to an overall reduction in the scale of prediction errors. Thus, we can see that the level of stochasticity in the dataset influences critical decision point detection.

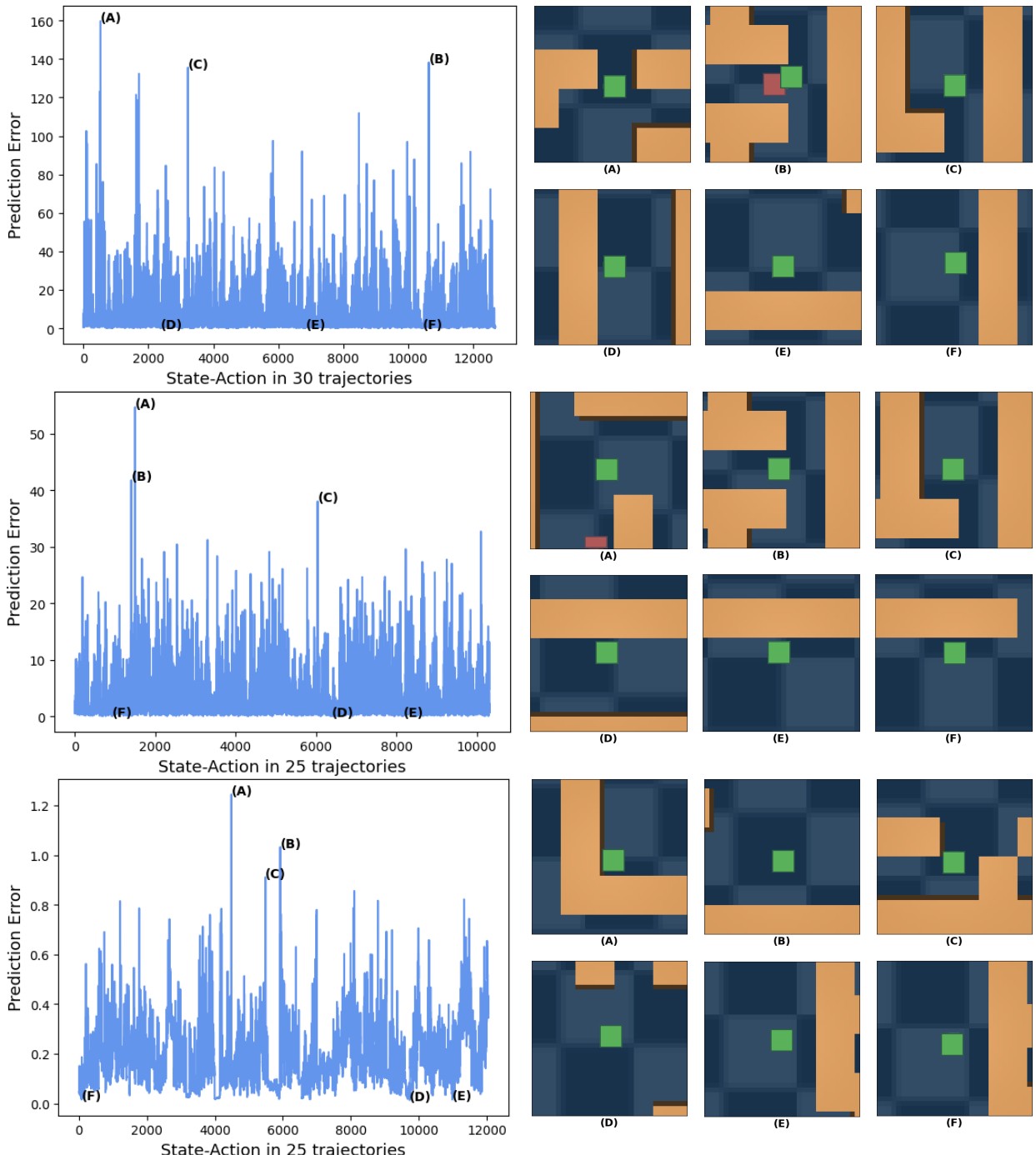

*Figure 15.* Visualization of prediction error of ICM in maze environment (**top**: trained with stochastic BC ($\sigma = 0.5$) dataset, **middle**: trained with stochastic BC ($\sigma = 0.75$) dataset, **bottom**: trained with random walk dataset). Note the same offline data that is used to train ICM was used to compute this prediction error. (A), (B) and (C) are the state-action pairs with the highest prediction error, while (D), (E) and (F) are the ones with the lowest.

## D. Visualization of Critical Decision Points in Complex Physics Simulation Tasks

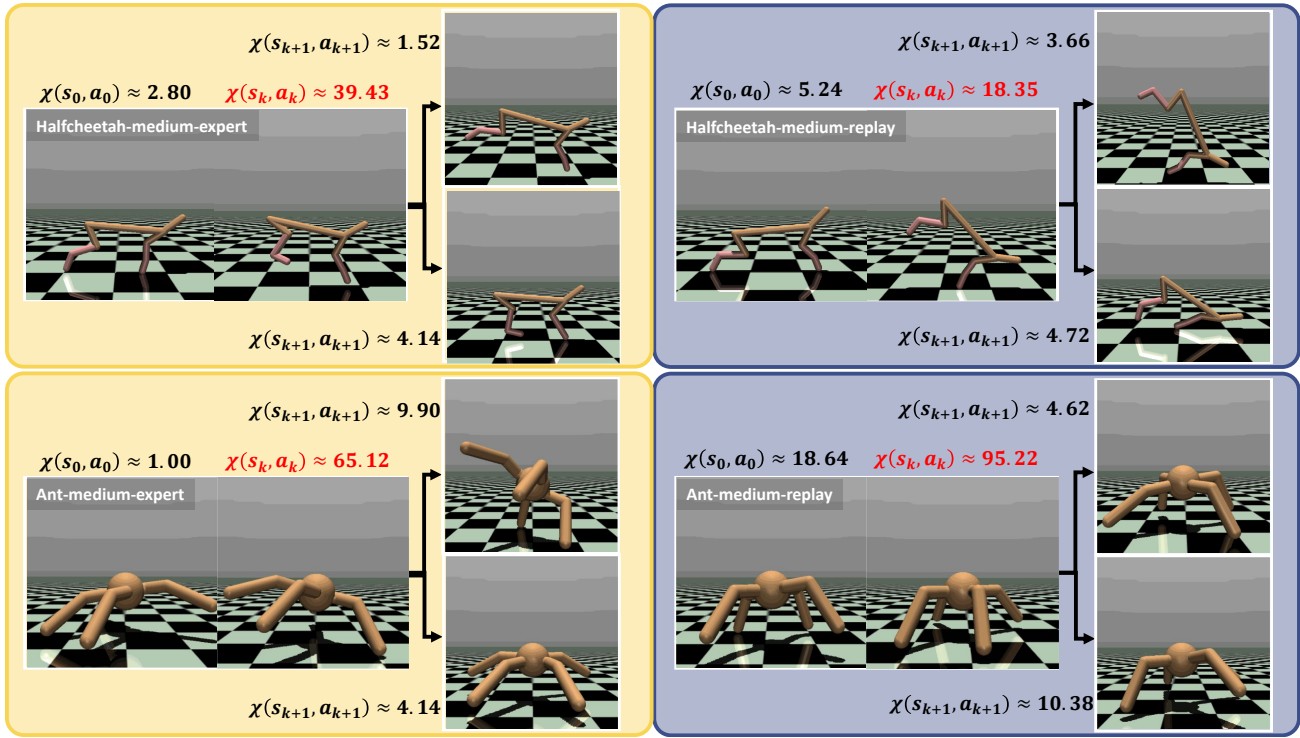

*Figure 16.* Visualization of critical decision points in MuJoCo (Todorov et al., 2012) environment (**top-left**: halfcheetah-medium-expert, **top-right**: halfcheetah-medium-replay, **bottom-left**: ant-medium-expert, **bottom-right**: ant-medium-replay)

We investigated whether meaningful decision points can be found in complex physics simulation tasks. We trained ICM using different offline datasets provided by D4RL (Fu et al., 2021) (halfcheetah-medium-expert, halfcheetah-medium-replay, ant-medium-expert, ant-medium-replay) to assess its ability to detect critical decision points. Figure 16 illustrates the presence of critical decision points in complex physics simulation tasks. For instance, the cheetah has the option of spreading its hind legs or lowering them to the ground, and the ant has the choice of flipping to the right or lowering themselves to the ground. However in completely random datasets (halfcheetah-random, ant-random), we were not able to find any meaningful decision points. Similar to Appendix C, it shows that the degree of stochasticity present in the offline dataset can influence critical decision point detection.

## E. Train/transfer Domain Similarity

To apply our method to downstream environments with significantly different overall layouts (e.g., a maze with a completely new structure or a larger-scale block stacking environment with more blocks in random positions), it is necessary to collect cropped image centered around the agent from task-agnostic demonstrations (visualizations of cropped images are available in Figure 2). This way, we can extract consistent structural information that can be extracted across multiple task-agnostic trajectories, which then can be used to detect state-action novelty based decision points even in downstream tasks with significantly different environment configurations. On the other hand, for environments where the overall layout remains consistent (e.g. the positions of manipulatable objects in the kitchen environment do not change during downstream task), we can use any state information that sufficiently describes the environment to apply our method (see Appendix H for environment details).

**Training Data** | **Downstream Tasks**

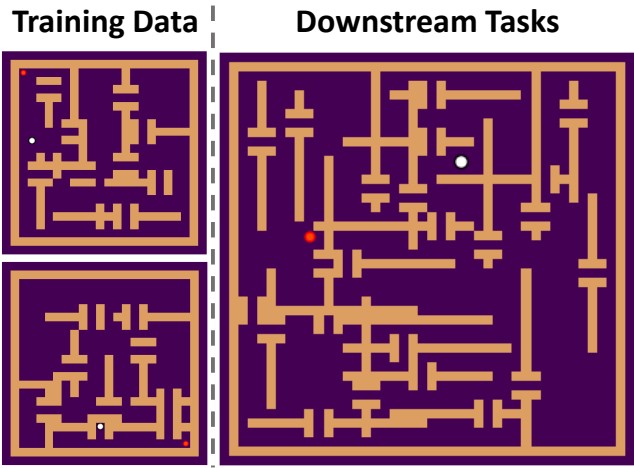

*Figure 17.* Examples of maze layouts used for training (left) and downstream task (right). The agent starts at the white point and aims to reach the red point.

## F. Task-Agnostic Dataset Size and Coverage

The number of task-agnostic trajectories collected for each environment were as follows: maze (85,000 trajectories), block stacking (37,000 trajectories), and kitchen (400 trajectories). Since we used image-based observations in the maze and block stacking environments, and the downstream tasks involve significant configuration changes (e.g., entirely new maze layouts or a larger-scale block stacking environment with more blocks in random positions), we require a larger set of task-agnostic demonstrations compared to the kitchen environment, where the overall layout remains consistent across tasks.

Since our goal in maze and block stacking environment is to solve downstream tasks with significantly different environment configuration, we do not assume that task-agnostic demonstrations fully cover the state space of the downstream tasks. However, it is important that they provide a good coverage of the observation space. Note we are using cropped image centered around the agent from task-agnostic demonstrations as observations. This way, we can extract consistent structural information that can be extracted across multiple task-agnostic trajectories, which then can be used to detect state-action novelty based decision points even in downstream tasks with significantly different environment configurations.

# G. Comparison to baseline methods on discrete state/action space

To provide an intuitive comparison between NBDI, Relative Novelty (Şimşek & Barto, 2004), and LOVE (Jiang et al., 2022) in discrete settings, we conducted a case study using a grid-based maze environment (Figure 18). This allowed us to directly visualize and compare termination points across methods in a controlled, discrete domain. To align with our task-agnostic setup, we collected diverse expert-level demonstrations from random start and goal positions, and used these datasets to extract skills and termination points.

Relative Novelty defines novelty at a given state as the ratio between the average novelty of future and past states, measured within a fixed-size sliding window (n_lag). A high value indicates that the agent transitions from a familiar region to a less familiar one. Following the original formulation, we only evaluated states with sufficient trajectory context on both sides of the window. As shown in the visualization, Relative Novelty is highly dependent on transition history, often identifying only a subset of bottleneck states.

LOVE uses a variational inference framework to extract skills based on the Minimum Description Length (MDL) principle in that its objective is to "effectively compress a sequence of data by factoring out common structure". While LOVE employs a variational inference framework to implement the MDL principle, we used Byte Pair Encoding (BPE) to extract skills in discrete setting as BPE is a specific formulation of MDL (Gallé, 2019) and it offers a more intuitive and interpretable formulation of compression. Termination points were visualized based on where the segmented skills terminated during 100 goal-reaching tasks. The results show that terminations vary significantly with the number of trajectories used to extract skills, as LOVE focuses on capturing common structure rather than consistent bottlenecks.

NBDI terminates skills based on both conditional action novelty and state novelty (Section 4.1). The visualization demonstrates that NBDI consistently identifies key bottleneck states and exhibits robustness to the number of trajectories collected in contrast to other methods. Moreover, states with high state novelty often correspond to regions that are rare or hard to reach within the task-agnostic dataset. By increasing the decision frequency in such unfamiliar states, NBDI promotes more effective exploration of the state space.

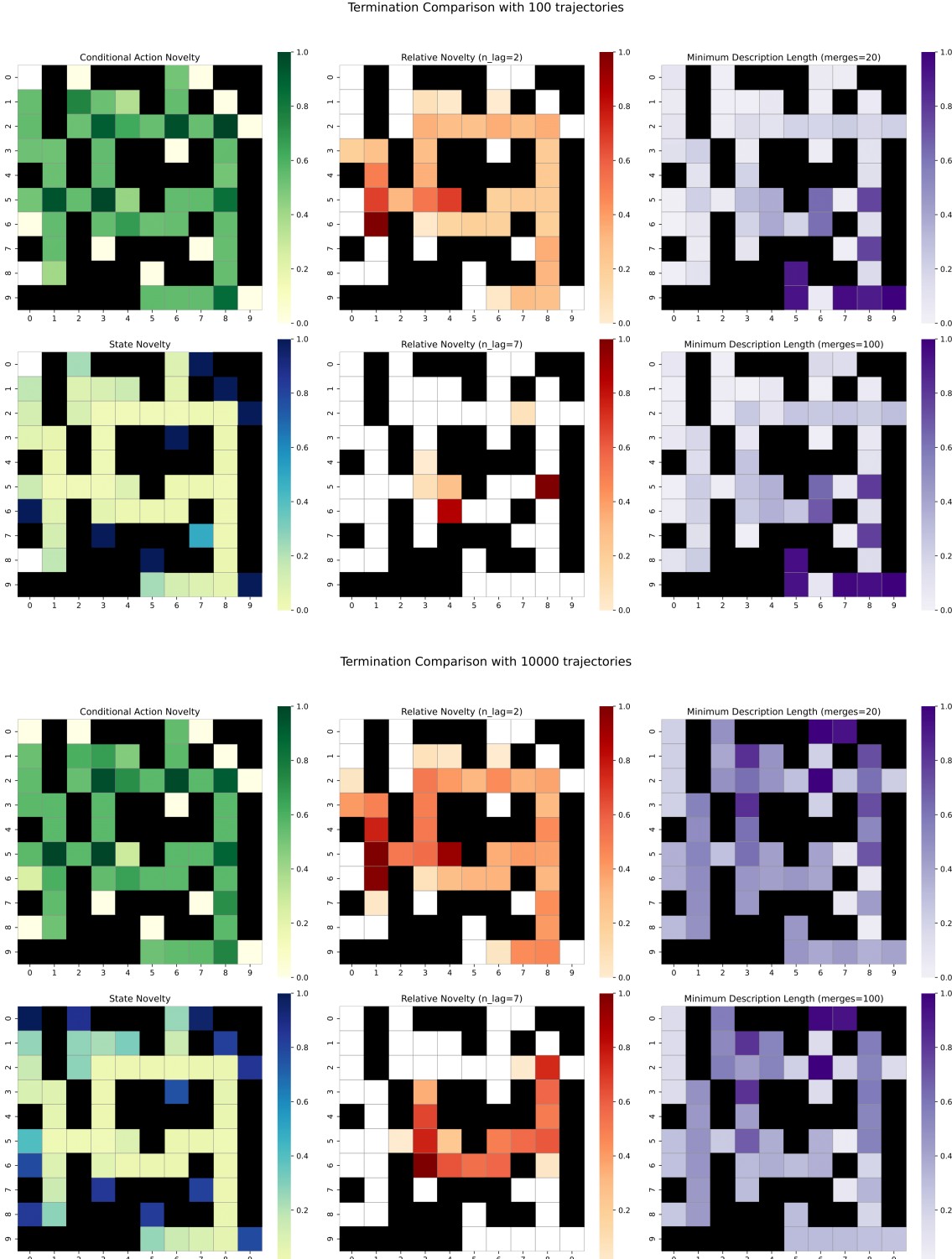

*Figure 18.* Visualization of skill termination points in a grid-based maze environment using NBDI (Conditional action novelty, State novelty), Relative Novelty, and LOVE (Minimum Description Length). Darker shades represent states with a relatively higher frequency of termination occurrences. Black cells are impassable wall regions in the environment. **Top:** termination points computed from 100 trajectories. **Bottom:** results from 10,000 trajectories.

# H. Data and Environment Details

We evaluate NBDI on three environments: one simulated navigation task (Maze navigation) and two simulated robotic manipulation tasks (Kitchen and Sparse block stacking). We employ the environment configuration and dataset provided by (Pertsch et al., 2021a) (Maze navigation and Sparse block stacking) and (Fu et al., 2021) (Kitchen) (CC BY 4.0). Note that task and environment setup differ between the training data and the downstream task, demonstrating the model's capacity to handle *unseen* downstream tasks.

**Kitchen Environment**    The kitchen environment is provided by the D4RL benchmark (Fu et al., 2021), featuring seven manipulable objects. The training trajectories consist of sequences of object manipulations. The downstream task of the agent involves performing an unseen sequence of four object manipulations.

The agent is tested by its ability to reassemble the skills learned from the training dataset to solve the downstream task.

- State space: 30-dimensional vector of the agent's joint velocities and the positions of the manipulatable objects

- Action space: 7-dimensional set for controlling robot joint velocities and a 2-dimensional set for gripper opening/closing degree

- Reward: one-time reward upon successfully completing any of the subtasks

**Maze Navigation**    The maze navigation environment is derived from the D4RL benchmark (Fu et al., 2021). During the collection of training data, a maze is generated randomly, and both the starting and goal positions are selected at random as well. The agent successfully reaches its goal in all of the collected trajectories.

In the downstream task, the maze layout is four times bigger than the one employed during training.

- State space: $(x, y)$-velocities and an image of local top-down view centered around the agent

- Action space: $(x, y)$-directions

- Reward: binary reward when the agent's position is close to the goal (computed using Euclidean distance)

**Sparse Block Stacking**    The sparse block stacking environment is created using the Mujoco physics engine. To gather training data, a hand-coded data collection policy interacts with a smaller environment with five blocks to stack as many blocks as possible.

In the downstream task, the agent's objective is to stack as many blocks as possible in a larger version of the environment with eleven blocks.

- State space: $(x, z)$-displacements for the robot and an image of local view centered around the agent

- Action space: 10-dimensional continuous symmetric gripper movements

- Reward: only rewarded for the height of the highest stacked blocks

**Differences to (Pertsch et al., 2021a).**    While (Pertsch et al., 2021a) employed a block stacking environment with dense rewards (the agent is rewarded based on the height of the stacked tower and for actions like picking up or lifting blocks), we evaluated our model and the baselines in a sparse block stacking environment, leading to different performance outcomes. In this setting, the agent is rewarded solely for the height of the tower it constructs, which increases the task complexity. Furthermore, there have been consistent reports indicating that performance in large maze environments displays a high level of sensitivity to random seeds, primarily due to the high stochasticity of the task. For the five different seeds that we used to compare the algorithms, we found the performance to be generally lower than what was previously reported, mainly due to the sensitivity to seeds.

# I. Implementation Details

Our codebase builds upon the released code of SPiRL (Pertsch et al., 2021a). Our code is available at: `https://github.com/ku-dmlab/NBDI`.

## I.1. Termination Improvement and Novelty

We present termination improvement and conditional action novelty in a simple $8 \times 8$ grid maze domain. As we mentioned in Appendix H, training data for the state-action novelty module is collected with diverse tasks. Thus, we set up the environment as follows: We randomly select one starting location and three goal locations to generate three trajectories for each goal location. The goal-reaching data collection policy randomly executes a discrete action, moving towards four different directions (left, right, forward, backward) while avoiding moving toward walls. The agent receives a binary reward when reaching the goal state.

We define an option $o = \langle \mathcal{I}, \pi, \beta \rangle$ where a deterministic policy $\pi$ follows the given trajectory, an initiation set $\mathcal{I} \subseteq \mathcal{S}$ defines all states that the policy visits, and a termination condition $\beta$ defines states where the option terminates (every option has a length of three). Each set of options, denoted as $\mathcal{O}_g$ for each goal $g = 1, 2, 3$, contains distinguishable options for each goal location. The frequencies of termination improvement occurrences in each state for each goal setting have been aggregated to generate Figure 3.

Using the trajectories collected from different goals, state novelty and conditional action novelty are simply computed as $\frac{1}{N(s)}$ and $\frac{N(s)}{N(s,a)}$ respectively. $N(s)$ represents the number of times a discrete state $s$ has been visited and $N(s,a)$ represents the number of times a discrete state-action pair has been used.

## I.2. State-Action Novelty Module

We use Intrinsic Curiosity Module (ICM) to calculate state-action novelty for both image-based and non-image-based observations. The feature encoder $\phi$, responsible for encoding a state $s_t$ into its corresponding features $\phi(s_t)$, is implemented differently for each environment. In the kitchen environment, it consists of a single fully-connected layer with a hidden dimension of 120. The both maze and sparse block stacking environments have three convolution layers with (4, 4) kernel sizes and (8, 16, 32) channels.

The forward dynamic model $f$ takes $a_t$ and $\phi(s_t)$ as inputs to predict the feature encoding of the state at time step $t + 1$. In the kitchen environment, the structure of the dynamic model is the same as its feature encoder. In the maze and sparse block stacking environment, a single fully-connected layer with hidden dimension 52 and 70 have been used, respectively.

The state-action novelty $\chi(s, a)$ is computed as the squared L2 distance between $\hat{\phi}(\phi(s_t), a_t)$ and $\phi(s_{t+1})$, representing the prediction error in the feature space. We employed the Adam optimizer with $\beta_1 = 0.9, \beta_2 = 0.999$ and a learning rate of $1e - 3$ to train the ICM. We found that state-actions within the top 1% prediction error percentile serve well as a critical decision points, and used the corresponding threshold to learn the termination distribution.

## I.3. BC+SAC and IQL+SAC

We trained flat offline learning methods, including Behavior Cloning (BC) and Implicit Q-Learning (IQL) (Kostrikov et al., 2021), using an offline dataset. In the downstream RL step, the trained policy $p(a|s)$ imposes a KL-divergence penalty on the SAC agent. In IQL, we set the expectile hyperparameter $\tau$ to 0.7 and the inverse temperature hyperparameter $\beta$ to 0.5 across all tasks.

$$J(\theta) = \mathbb{E}_\pi \left[ \sum_{t \in \mathcal{T}} r(s_t, a_t) - \alpha D_{\mathrm{KL}} \left( \pi(a_t|s_t), p(a_t|s_t) \right) \right] \tag{6}$$

## I.4. LOVE and Relative novelty

Since LOVE (Jiang et al., 2022) requires a discrete action space for option learning, we implemented some modifications. In maze environment, we discretized the continous action space into 8 discrete actions. In the kitchen environment, we modified the action decoder to handle continuous distribution.

*Table 6.* State-Action Novelty Module Hyperparameters

| HYPERAPARAMETER | VALUE |
|---|---|
| BATCH SIZE | 150 |
| OPTIMIZER | ADAM($\beta_1 = 0.9, \beta_2 = 0.999, lr = 1e-3$) |
| LOSS WEIGHT $\beta$ | 0.2 |
| SCALING FACTOR $\eta$ | 1.0 |
| KITCHEN | |
|   FEATURE ENCODER | |
|     HIDDEN DIM | 120 |
|   FORWARD DYNAMIC MODEL | |
|     HIDDEN DIM | 120 |
| MAZE | |
|   FEATURE ENCODER | |
|     KERNEL SIZE | (4, 4) |
|     CHANNELS | 8, 16, 32 |
|   FORWARD DYNAMIC MODEL | |
|     HIDDEN DIM | 52 |
| SPARSE BLOCK STACKING | |
|   FEATURE ENCODER | |
|     KERNEL SIZE | (4, 4) |
|     CHANNELS | 8, 16, 32 |
|   FORWARD DYNAMIC MODEL | |
|     HIDDEN DIM | 70 |

Since relative novelty (Şimşek & Barto, 2004) used pseudo counts to measure state novelty in discrete environments, we could not apply this method to continuous environments straightforwardly. Furthermore, since this technique requires environments that support backward transitions for computing the relative novelty score, its direct application in general environments seems challenging. Thus, to align with the author's intuition that target states should exhibit higher novelty scores compared to non-target states, we utilized our state novelty module $\chi(s)$ to measure the state novelty, and trained a neural network with offline datasets to estimate the relative novelty score of states.

### I.5. Experiments Compute Resources

Each experiment was conducted on a single CPU (Intel Xeon Gold 6330) with 256GB of RAM and a single GPU (NVIDIA RTX 3090). Each training session took about 36 hours (12 hours for prior learning, 24 hours for downstream learning), utilizing approximately 30% of the RAM and 25% of the GPU memory. We implemented all RL algorithms using PyTorch v1.3 and executed them on Ubuntu 22.04.04 LTS.

