# OpenReview forum: "NBDI: A Simple and Effective Termination Condition for Skill Extraction from Task-Agnostic Demonstrations"
_ICML.cc/2025/Conference — ICML 2025 poster_

### Official Review · Reviewer_Qf8T · 2025-03-09

**Overall Recommendation:** 4

**Summary:**

The paper considers the problem of learning the termination conditions of the Option framework from task-agnostic demonstrations (demonstrations that are either exploratory or for other tasks). The key idea of the paper is to use state-action novelty to find states where termination will likely be reasonable to choose alternative skills: when a skill is finished or a decision-making should be made at an important point, the error of predicting the next state is lower. The paper's approach enables skills to be variable length based on the learned termination condition. The experiments show that the proposed approach improves performance on learning in unseen tasks by utilizing the termination condition to make essential decisions at appropriate states. The authors also conduct comprehensive benchmark tests against other termination condition learning approaches as well as ablations of its components. The results strongly support the claims made by the paper. In general, the paper presentation is very clear and easy to follow. Although the novelty of the paper is somewhat limited (by utilizing a previously known method of curating novelty to identify termination conditions), I believe the paper presents some meaningful contribution in the novel idea of applying novelty measure in the option framework, and the results show the benefit of doing so.

**Claims And Evidence:**

Yes.

**Essential References Not Discussed:**

Not that I am aware of

**Experimental Designs Or Analyses:**

Yes, the experimental design is sound and through in comparing with benchmarks and ablations of the proposed approach.

**Methods And Evaluation Criteria:**

The idea of using novelty to detect termination conditions for options framework makes sense. The evaluation is done to see if the better termination conditions help options framework to adapt to unseen tasks faster and achieve better performance.
1. One concern I do have is for Section 6.6 and Figure 8. Why is "mean value of state-action novelty for 25 trajs" a good metric to state "proposed approach is robust to various number of parameters or size of datasets"? To me, learning the mean value is the easiest thing to do for ML, and of course with an i.i.d. sampled sub dataset or a smaller neural network it can still learn it. The key of ML is to learn the variances among data (some data corresponds to a high predicted value vs. some data corresponds to a low predicted value). Please clarify if I misunderstood, otherwise I believe the experiment for Section 6.6 should be redone with a different metric.

**Other Comments Or Suggestions:**

The Figure 2 should be placed later in the paper as it is no introduced until Section 6.

**Other Strengths And Weaknesses:**

Although I find the paper overall well-written, I do have a few suggestions for the structure of the paper. Currently, the technical details of Section 5.1 and 5.2 are largely in supplementary, which I believe hinders the ability for readers that are less familiar with the topic to appreciate the work. The learning of the novelty function \Chi(s,a), and the significant novelty criteria to determine termination, \beta, are important details to understand the reproduce the approach. As such, I believe moving them into the main paper would be beneficial, and for the page limit's sake, some contents of Section 6.1 could be moved to supplementary, as the illustration of Figure 2a and 2b seem very similar. Similarly, Figure 3 seems hard to interpret without reading Appendix G.1, and I recommend at least providing a brief explanation of how the figure is generated.

**Questions For Authors:**

1. According to Section 5.1, z is the embedding of the s-a pairs and \beta from time t to t+H-1. It is unclear to me how this information is available during execution time - how can one predict the future trajectory in order to calculate z?
2. According to line 318 (right column), "a large set of task-agnostic agent experiences is collected" - how large is this dataset? Is it assumed that the demonstration set covers the state space well?

**Relation To Broader Scientific Literature:**

The contribution of the paper is to relax previous option framework's fixed-length skills to be variant-length skills according to the learned termination function.

**Theoretical Claims:**

The paper does not have theoretical claims.

---

> ### Author Rebuttal · Authors · 2025-04-01
>
> Thank you for your review and constructive suggestions. We address your questions below.
>
> **Q1: Details in skill termination and execution**
>
> Thank you for your feedback. To clarify, the variable-length skill embedding space $z$ is learned **offline** during the skill extraction phase using a deep latent variable model. During this phase, the low-level policy (i.e., the skill decoder in Figure 12) receives a randomly sampled experience from the task-agnostic demonstration, along with a termination condition vector $\beta$ provided by the pre-trained state-action novelty module. The latent model is trained to reconstruct the corresponding action sequence and its length (i.e., the termination point) by maximizing the evidence lower bound (ELBO) (Appendix B.1).
>
> During the execution phase (or when solving downstream tasks), the pre-trained low-level policy (i.e., the skill decoder) takes as input a latent skill vector $z$ from the high-level policy and reconstructs action and termination signal $\beta$ at each step, without needing to observe or predict future states or actions (Appendix B.2). Thus, while future trajectory segments are used during skill extraction to learn variable-length skill embeddings and terminations, they are **not required at execution time**, ensuring our approach remains practical and deployable.
>
> **Q2: Task-Agnostic Dataset Size and Coverage**
>
> Thank you for your feedback. The number of task-agnostic trajectories collected for each environment were as follows: maze (85,000 trajectories), block stacking (37,000 trajectories), and kitchen (400 trajectories). Since we used **image-based observations** in the maze and block stacking environments, and the downstream tasks involve significant configuration changes (e.g., entirely new maze layouts or a larger-scale block stacking environment with more blocks in random positions), we require a larger set of task-agnostic demonstrations compared to the kitchen environment, where the overall layout remains consistent across tasks (Appendix F).
>
> Since our goal in maze and block stacking environment is to solve downstream tasks with significantly different environment configuration, we do not assume that task-agnostic demonstrations fully cover the **state space** of the downstream tasks. However, it is important that they provide a good coverage of the **observation space**. Note we are using cropped image centered around the agent from task-agnostic demonstrations as observations. This way, we can extract consistent structural information that can be extracted across multiple task-agnostic trajectories, which then can be used to detect state-action novelty based decision points even in downstream tasks with significantly different environment configurations (Appendix F).
>
> **Q3: Impact of model capacity and dataset usage**
>
> Thank you for the insightful feedback regarding the experiments. We agree with the reviewer that simply reporting the mean and standard deviation of state-action novelty across 25 trajectories may not sufficiently demonstrate the robustness of our method with respect to model capacity or dataset size. The concern that machine learning models can trivially capture mean values under i.i.d. sampling is valid, and we appreciate the opportunity to clarify our intent.
>
> To address this, we revised the analysis in Figure 8 to present box plots that visualize the full distribution of state-action novelty scores across 25 sampled trajectories. The revised version of Figure 8 is available at: https://imgur.com/a/9gQ8IrU. This allows us to assess the variance, range, and consistency of state-action novelty estimates across different network sizes (left) and dataset usage levels (right). From the revised plots, it can be observed that the spread (variance, median and interquartile range) of state-action novelty remains relatively stable across different model sizes and dataset usage.
>
> **Q4: Paper structure rearrangement**
>
> Thank you for your thoughtful suggestions regarding the structure of the paper. We agree that the current presentation of Sections 5.1 and 5.2 may limit accessibility for readers who are less familiar with skill extraction from task-agnostic demonstrations. To address this, we will incorporate concise summaries of Appendix B.1 and B.2 into Sections 5.1 and 5.2. Additionally, we will expand the caption of Figure 3 to briefly explain the visualization procedure used to generate the figure. This added context will help readers better interpret the figure and understand its significance.
>
> To accommodate these changes within the page limit, we will move Figure 2(b) and its accompanying explanation (currently in Section 6.1) to the appendix.  Furthermore, we will relocate Figure 2 to Section 6, where it is first referenced, to improve the flow of the paper.

---

> > ### Comment · Reviewer_Qf8T · 2025-04-02
> >
> > Thank you to the authors for providing detailed answers to my questions.
> >
> > Q1: clearly answered.
> >
> > Q2: I think it is important to provide an operational definition for "a good coverage of the observation space", such that future works could have a guidance of how much data is required to replicate the success of NBDI on other domains.
> >
> > Q3: clearly addressed with the new evidence.
> >
> > Q4: clearly answered.
> >
> > As the authors have addressed my main concerns, I increase my overall score from 3 to 4.

---

### Official Review · Reviewer_bZqs · 2025-03-13

**Overall Recommendation:** 4

**Summary:**

This paper presents  NBDI (Novelty-based Decision Point Identification), a state-action novelty-based decision point identification method that allows an agent to learn terminated skills from task-agnostic demonstrations. The key advancement presented in this work includes the mechanism for determining critical decision points, allowing variable-length skills to be learned. The authors present a cohesive methodology section followed by an abundance of results. Overall, the work concludes with several key advancements and interesting insights, displaying the importance of the formulation of the state-action novelty-based termination module.


### Post-Rebuttal
Thank you for your responses regarding the diversity of the task demonstration set and the average length of skills learned. Please include this in the updated manuscript.

**Claims And Evidence:**

Yes, the claims are supported by an abundance of evidence.

**Essential References Not Discussed:**

None

**Experimental Designs Or Analyses:**

Yes, the validity of the experimental design was checked thoroughly.

**Methods And Evaluation Criteria:**

Yes, the paper has sufficient evaluation for the problem at hand.

**Other Comments Or Suggestions:**

- Could you comment on how diverse the initial task demonstration set can be (or needs to be)?

**Other Strengths And Weaknesses:**

Strengths:
+ The paper contains abundant results and support for the proposed method.
+ There are several intuitive explanations that help with understanding the paper.


Weaknesses:
- Due to the amount of material within the paper, many important details are found in the Appendix. The paper could be improved with longer figure captions.

**Questions For Authors:**

- Could you provide further details regarding the average length of skills learned?

**Relation To Broader Scientific Literature:**

Yes, the work is compared to several related works in Section 2 and compared against several works in Section 6.

**Theoretical Claims:**

N/A

---

> ### Author Rebuttal · Authors · 2025-04-01
>
> Thank you for your valuable feedback. Please find our responses to your concerns below.
>
> **Q1: Diversity of the task demonstration set**
>
> Thank you for your insightful suggestion. In our experiments, we found that the initial task-agnostic demonstration set needs to be diverse enough to cover the **observation space** (i.e., cropped images centered around the agent (Figure 2)), rather than fully covering the **state space** of downstream tasks. This diversity is especially important in environments like maze and block stacking, where downstream tasks involve significant changes in configuration (e.g., unseen maze layouts or random block arrangements in a larger-scale block stacking environment). To ensure sufficient diversity, we used 85,000 trajectories in the maze environment and 37,000 trajectories in the block stacking environment. In contrast, the kitchen environment, where the scene layout is consistent across tasks, required only 400 trajectories.
>
> **Q2: Details in the average length of skills learned**
>
> Thank you for your feedback. In Appendix A.4, we explored the impact of fixed skill trajectory lengths by evaluating whether NBDI still outperforms SPiRL [1] when SPiRL is configured to use the average skill length observed in NBDI (referred to as SPiRL (avg-fixed)). We found that NBDI consistently outperforms SPiRL even when SPiRL uses these average fixed skill lengths. Interestingly, SPiRL (avg-fixed) performed worse than SPiRL with a fixed skill length of 10 in the block stacking environment, but better in the maze environment. However, across all fixed skill lengths tested (ranging from 10 to 30), there was no instance where SPiRL outperformed NBDI. These findings support our claim that NBDI can effectively leverage critical decision points in the environment compared to fixed length approaches.
>
> Furthermore, in Table 2 of Appendix A.4, we report the skill lengths of high-level policy for each environment during downstream reinforcement learning. The results show that our method maximizes temporal abstraction by frequently executing longer skills (lengths 21–30). At the same time, it retains the flexibility to terminate earlier (lengths 1–10) when critical decision points are encountered. This adaptability enables more efficient exploration of the state space and enhancing the transfer of knowledge across various tasks.
>
> **Q3: More informative figure captions**
>
> Thank you for your suggestion. We will revise the captions for Figure 1 and Figure 2 to provide additional context about the tasks and domains depicted. Additionally, we will expand the caption for Figure 3 to briefly explain the visualization procedure, thereby helping readers better interpret the figures and their significance.
>
> [1] P., Karl, et al. "Accelerating reinforcement learning with learned skill priors." Conference on robot learning. PMLR, 2021.

---

> > ### Comment · Reviewer_bZqs · 2025-04-02
> >
> > Thank you for your responses regarding the diversity of the task demonstration set and the average length of skills learned.

---

### Official Review · Reviewer_M4gG · 2025-03-14

**Overall Recommendation:** 4

**Summary:**

This paper introduces NBDI (Novelty-based Decision Point Identification), a novel approach for learning termination conditions to extract skills from task-agnostic demonstrations. The method consists of using state-action novelty to identify critical decision points where skills should terminate, allowing for variable-length skill execution. The method leverages an intrinsic curiosity module (ICM) to estimate novelty, considering state-action pairs with high novelty scores as potential decision points. NBDI is evaluated across multiple environments (maze navigation and robotic manipulation tasks) and consistently outperform fixed-length skill extraction methods like SPiRL. The authors demonstrate that by executing variable-length skills that terminate at meaningful decision points, agents can explore more efficiently and better transfer knowledge across different tasks.

**Claims And Evidence:**

The claims are generally well-supported by evidence. The authors demonstrate NBDI's greater performance compared to fixed-length skill approaches (like SPiRL) across multiple environments. The claim that state-action novelty can identify meaningful decision points is supported by visualizations showing that high novelty points correspond to intuitive decision points (e.g., crossroads in mazes, and completed subtasks in manipulation tasks).

The authors are transparent about limitations regarding sensitivity to dataset quality. They provide experiments (Section 6.5) and visualizations (Appendix C) to demonstrate how varying levels of stochasticity in demonstrations affect decision points detection.

**Essential References Not Discussed:**

I am not aware of any essential reference not discussed in this work.

**Experimental Designs Or Analyses:**

The analyses appear sound. The authors conduct experiments across multiple seeds (five different seeds per experiment) and report confidence intervals. They compare against ablations, fixed-length skill methods (SPiRL), other variable-length skill approaches (LOVE, relative novelty), and even algorithms that do not leverage temporal abstractions (flat RL methods). The analyses of (1) how dataset quality affects performance and (2) where the decision points are placed in maze environments both add valuable insights.

**Methods And Evaluation Criteria:**

The proposed methods and evaluation criteria are appropriate. The authors evaluate NBDI on two navigation tasks and two robotic manipulation tasks, providing diverse testing environments with continuous state and action spaces. They compare against multiple relevant baselines (SAC, BC+SAC, IQL+SAC, SPiRL, and other variable-length skill methods).

The metrics used (success rate for navigation, number of stacked blocks for manipulation, and number of completed subtasks for kitchen environment) are suitable for the respective tasks. The ablation studies (with "NBDI-x" baselines, or in Appendix A) isolate contributions from different novelty measures and design choices.

**Other Comments Or Suggestions:**

The paper would benefit from clearer explanation of the termination condition implementation in Section 5.1.

**Other Strengths And Weaknesses:**

Strengths:

- The approach is simple and clear. The theoretical motivation linking state-action novelty to termination improvement is well-justified.
- Visualizations nicely illustrate how the detected decision points align with intuitive critical points in the environment.

Weaknesses:

- The novelty threshold is a critical hyperparameter requiring environment-specific tuning, which may limit generalizability.
- The effectiveness appears sensitive to the quality of demonstration data, potentially limiting applicability with highly stochastic or suboptimal demonstrations.

**Questions For Authors:**

1. Given the significant variation in novelty thresholds across environments (0.3 for kitchen vs. 50 for maze tasks), have you explored adaptive threshold setting approaches? For instance, would it be effective to set a coefficient $\alpha$ as a hyperparameter, track the maximum novelty score $m$ observed during training, and consider all state-actions whose novelty is higher than $\alpha m$ as critical decision points?
2. How would you expect NBDI to compare against LOVE on discrete action environments, and against "relative novelty" approaches on discrete state spaces? This could help clarify the method's advantages across different types of environments.

**Relation To Broader Scientific Literature:**

The paper positions itself well within the literature on skill-based RL, particularly regarding skill extraction and termination condition learning. It builds upon the option framework (Sutton, 1998), and SPiRL (Pertsch et al., 2021), a recent fixed-length skill extraction method.

The authors acknowledge prior work on identifying sub-goals through novelty (Şimşek & Barto, 2004), but extend this to continuous state-action spaces. The work also relates to research on curiosity-driven exploration (Pathak et al., 2019) by repurposing novelty estimation for termination conditions rather than exploration bonuses.

**Theoretical Claims:**

Not applicable (as no new theoretical claims)

---

> ### Author Rebuttal · Authors · 2025-04-01
>
> Thank you for your thoughtful comments. Please find the responses to your questions below.
>
> **Q1: Details in implementation of termination conditions**
>
> Thank you for the suggestion. To improve clarity, we will revise Line 232 (second column) to include a more detailed explanation of the termination condition implementation, which is currently described in Appendix B.1. The revised sentence will read:  “During the training of the low-level policy, the deep latent variable model receives a randomly sampled experience from the training dataset, along with a termination condition vector $\beta$ provided by the state-action novelty module. The model is trained to reconstruct the corresponding action sequence and its length (i.e., the termination point) by maximizing the evidence lower bound (ELBO).”
>
> **Q2: Adaptive thresholds for environments**
>
> Thank you for your insightful suggestion. In practice, the thresholds we tuned through experiments (Appendix A.1) approximately correspond to the 97th percentile of the novelty values computed over task-agnostic demonstrations—e.g., kitchen = 0.3 (97.47th percentile), maze = 50 (96.86th percentile), and block stacking = 40 (96.12th percentile). We will include this percentile-based guidance in the paper to help readers more easily apply our method across new environments.
>
> **Q3: Comparison NBDI to other methods on discrete state/action space**
>
> Thank you for your valuable suggestion. To provide an intuitive comparison between NBDI, Relative Novelty[1], and LOVE[2] in discrete settings, we conducted a case study using a grid-based maze environment. This allowed us to directly visualize and compare termination points across methods in a controlled, discrete domain. To align with our task-agnostic setup, we collected diverse expert-level demonstrations from random start and goal positions, and used these datasets to extract skills and termination points. The figure is available at: https://imgur.com/a/aRFrZ30.
>
> [1] defines novelty at a given state as the ratio between the average novelty of future and past states, measured within a fixed-size sliding window (n_lag). A high value indicates that the agent transitions from a familiar region to a less familiar one. Following the original formulation, we only evaluated states with sufficient trajectory context on both sides of the window. As shown in the visualization, Relative Novelty is highly dependent on transition history, often identifying only a subset of bottleneck states.
>
> [2] uses a variational inference framework to extract skills based on the Minimum Description Length (MDL) principle in that its objective is to “effectively compress a sequence of data by factoring out common structure”. While [2] employs a variational inference framework to implement the MDL principle, we used Byte Pair Encoding (BPE) to extract skills in discrete setting as BPE is a specific formulation of MDL [3] and it offers a more intuitive and interpretable formulation of compression. Termination points were visualized based on where the segmented skills terminated during 100 goal-reaching tasks. The results show that terminations vary significantly with the number of trajectories used to extract skills, as [2] focuses on capturing common structure rather than consistent bottlenecks.
>
> NBDI terminates skills based on both conditional action novelty and state novelty (Section 4.1). The visualization demonstrates that NBDI consistently identifies key bottleneck states and exhibits robustness to the number of trajectories collected in contrast to other methods. Moreover, states with high state novelty often correspond to regions that are rare or hard to reach within the task-agnostic dataset. By increasing the decision frequency in such unfamiliar states, NBDI promotes more effective exploration of the state space.
>
>
> [1] Ş., Özgür, and A. Barto. "Using relative novelty to identify useful temporal abstractions in reinforcement learning." Proceedings of the twenty-first international conference on Machine learning. 2004.
>
> [2] J., Yiding, et al. "Learning options via compression." Advances in Neural Information Processing Systems 35 (2022): 21184-21199.
>
> [3] G., Matthias. "Investigating the effectiveness of BPE: The power of shorter sequences." Proceedings of the 2019 conference on empirical methods in natural language processing and the 9th international joint conference on natural language processing (EMNLP-IJCNLP). 2019.

---

> > ### Comment · Reviewer_M4gG · 2025-04-03
> >
> > Thank you very much for your responses. They are all very clear and address all the interrogations I had. Furthermore, I completely agree with the suggested additional content and experiment.

---

### Official Review · Reviewer_A7Qu · 2025-03-18

**Overall Recommendation:** 2

**Summary:**

The paper is on the topic of skill learning in reinforcement learning. Its contribution is on the subject of when skills should be terminated. The authors build on the existing work on the literature on skill learning, where the learned skills are executed for a fixed number of time steps. Here the authors add the flexibility of terminating the skills at other times. Specifically, they focus on identifying states that would serve as particularly useful points of termination based on the novelty of the state and the stat-action pair. The authors present empirical results in several domains.

**Claims And Evidence:**

The main claims are as follows:

(1) The proposed skill termination condition for task-agnostic demonstrations remains effective even when the environment configuration of complex, long-horizon downstream tasks undergo significant changes.

(2) The paper presents "a novel finding in reinforcement learning, which is the identification of state-action novelty-based critical decision points."

(3)  Executing terminated skills, based on state-action novelty, can substantially enhance policy learning in both robot manipulation and navigation tasks.

It is not clear how well the paper supports claim (1). There is not a clear description of the test environments and how they change in future tasks.

The second claim is confusing because "identification" of critical decision points is not a "finding".

For the third claim, there is some supporting evidence in the paper. However, it must be noted that the experimental evaluation does not consider a broad range of skill-based approaches to policy learning.

**Essential References Not Discussed:**

The paper proposes a termination condition for skills, which could in potential be useful with many existing approaches to skill discovery that do not identify termination conditions for the discovered skills and instead executes them for fixed time lengths. But the paper examines only one such approach (SPiRL). A few other approaches are mentioned in the paper but not included in the experimental evaluation. I suggest that the authors include a broader discussion of existing approaches to skill learning for which their approach may be useful and that they include some of these methods in their experimental evaluation.

**Experimental Designs Or Analyses:**

No particular problems I have noticed.

**Methods And Evaluation Criteria:**

The environments and the tests that are carried out make sense for the problem at hand. However, ideally a broader set of skill-based methods would be used in the evaluation. Currently, only a single approach is tested (SPiRL)

**Other Comments Or Suggestions:**

Figure 1 -- The figure shows a number of images from the kitchen environment but without sufficient context. The reader needs a better understanding of the domain in order to get meaningful information from these images.

Figure 2 -- The last figure here has low resolution. More importantly, in both figures showing states from the environment, it is not fully clear what these figures are showing. The domains, and their visual depiction, needs to be explained to the reader.

Line 198 -- "where a subtask has been completed" -- What is a "subtask"? How is it defined? Additionally, this domain needs to be explained in more detail.

Figure 3 -- Needs a larger font size on the side bar showing the color scale.

Line 185-190, second column: "When the skills (or options) are discovered from diverse trajectories (e.g., trajectories gathered from a diverse set of goals), termination improvement is typically observed in states where a multitude of actions have been executed, such as crossroads." It is not clear what the basis is for this observation.

Line 264 - 268, column 1. Here a discussion on the agent policy would be useful. Does the policy matter? Are we making any assumptions about the agent policy when trajectories are collected. Would there be a difference if the agent is acting optimally within the current task versus acting randomly?

Figure 5. Why is the success rate so low in the two mazes? In sparse block stacking, what is the maximum number of stacked blocks possible?

Line 325 -- The problem should be specified by using the reward function not in terms of what the agent needs to do. How specially is the agent being rewarded?

Line 318: "A large set of task-agnostic agent experiences is collected from each environment" Please state what the agent policy was in these environments. Was it the optimal policy for the task? Something else?

There is very little description of the environments in the main paper. A large part of the evidence provided against the claims in the paper comes from empirical evaluation. Therefore the structure of the environments is important in understanding the strengths and weaknesses of the proposed approach. Ideally the main paper would provide sufficient high-level description of the environments, and this is currently not the case. These include, for example, the states, actions, reward function, and some explanation of the domain dynamics, for example, the level of stochasticity.

For baseline skill-based algorithms with fixed execution length, it would be useful to explore the impact of the skill trajectory length on the results.

**Other Strengths And Weaknesses:**

The main idea in the paper -- that skills should be terminated at points where renewed decision making may be useful -- is I think a good one. It may be useful in combination with many different ways of discovering skill policies.

The writing can be more organised and clear. I have given some specific comments in the next section, which I hope the authors will find useful.

**Questions For Authors:**

Do you think your proposed approach would be useful in conjunction with other existing methods of determining skill policies? Which ones? Have you done any analysis of the use of the proposed skill termination with different ways of setting skill policies?

**Relation To Broader Scientific Literature:**

The paper implements an intuitive idea for skill termination in reinforcement learning. This proposed approach to skill termination is an alternative to executing skills for a pre-determined fixed period of time.

There are a large number of approaches in the literature to skill discovery -- here I am using the term "skill" broadly to denote a behaviours that can take multiple time steps to execute. Many of these approaches identify termination conditions explicitly, which means that these skills terminate in accordance with their termination conditions rather than after executing for a fixed period of time steps. So the ideas in this paper will not necessarily be relevant for those approaches.

However, there are also some existing methods that learn skills with no explicit termination conditions; with these methods, the skills are executed for a fixed amount of time. The proposed approach can be useful for such skills.

**Theoretical Claims:**

N/A

---

> ### Author Rebuttal · Authors · 2025-04-01
>
> We thank the reviewer for the thorough and constructive comments. We hope we can address your concerns below.
>
> **Q1: Description of the train/transfer environments**
>
> In Appendix E, we introduced the train/transfer domain similarity metrics for the environments used in our experiments. To demonstrate the effectiveness of our method in challenging downstream tasks involving significant configuration changes, we tested on a maze with an unseen layout and a larger-scale block stacking environment with more blocks in random positions —both of which differ significantly from the task-agnostic offline datasets. The significant performance improvements reported in Section 6 clearly demonstrate the effectiveness of our method.
>
> **Q2: Confusing expression**
>
> We apologize for the confusion caused by our wording. We agree that the term "finding" may have been misleading in this context. We will revise the word to “a novel method”.
>
> **Q3: Applying NBDI to other skill-based approaches**
>
> Thank you for your constructive suggestion regarding the experiments. Our work focuses on extracting meaningful termination points from task-agnostic demonstrations. To explore the broader applicability of our approach, we investigated its compatibility with a skill-based approach that leverages task-agnostic demonstrations to solve challenging long-horizon, sparse-reward meta-RL tasks [1]. Specifically, we applied our method during the skill extraction phase of [1] to learn variable-length skills in place of fixed-length skills.
>
> In the maze environment, we randomly sampled 10 goal locations for meta-training. The table below compares the success rate of meta-policies trained with [1] (fixed-length skills) and our method during the meta-training phase across episodes. The result shows that the extracted variable-length skills allows the meta-policy to better promote knowledge transfer between different tasks, helping the meta-policy in combining the skills to complete complex tasks. We report mean success rates on the 10 meta-training tasks across 3 different seeds with standard errors.
> | |ep200|ep400|ep600|ep800|ep1000|
> |---|---|---|---|---|---|
> |[1]|0.193$\pm$0.030|0.329$\pm$0.037|0.501$\pm$0.049|0.436$\pm$0.048|0.582$\pm$0.037|
> |[1]+NBDI|**0.341**$\pm$0.035|**0.554**$\pm$0.037|**0.737**$\pm$0.028|**0.828**$\pm$0.028|**0.896**$\pm$0.025|
>
> Furthermore, during the target task learning phase, the meta-policy learned through our approach leads to **significantly better sample efficiency** on the unseen target task. These results indicate that our approach can be effectively integrated with a broader class of skill-based methods that leverage task-agnostic demonstrations (3 seeds).
> | |ep20|ep100|ep300|ep500|
> |---|---|---|---|---|
> |[1]|0.143$\pm$0.063|0.593$\pm$0.191|0.667$\pm$0.193|0.990$\pm$0.006|
> |[1]+NBDI|**0.560**$\pm$0.121|**0.960**$\pm$0.021|**0.980**$\pm$0.011|**0.993**$\pm$0.003|
>
> **Q4: The trajectory collecting policy**
>
> As our work primarily focuses on learning termination conditions from task-agnostic demonstrations, we will revise Line 264–268 and 318 to clarify that we assume access to **task-agnostic, expert-level demonstrations**. In Section 6.5, we analyzed the impact of dataset quality on our method’s performance. The results indicate that the level of stochasticity and the suboptimality of the dataset influence the effectiveness of our approach, which remains a limitation of our work.
>
> **Q5: Skill length impact on baselines**
>
> Due to space limitations, we kindly refer you to our response to **Reviewer bZqs’s Q2**, where we provide a detailed discussion.
>
> **Q6: Details in environment settings**
>
> We apologize for not including information about the environments (Appendix E, F) in the main paper. In the kitchen environment, the agent completes an unseen sequence of object manipulation subtasks, like opening the cabinet or turning on the oven. In the maze environment, it receives a binary reward for reaching the goal. In the sparse block stacking environment, the agent is rewarded based on the height of successfully stacked blocks at the end of the episode. We will include a brief introduction to the environments in the experiment section.
>
> **Q7: Other comments and suggestions**
>
> **(Line 185-190)** We apologize for the confusion. Since the statement in Line 185–190 is based on Figure 3, we will move it to the paragraph where Figure 3 is discussed in detail.
>
> **(Figure 5)** Figure 5(a) and 5(b) illustrate the performance of the baselines and our method on a challenging downstream maze task, which involves reaching the goal in a maze with an unseen layout (Appendix E). Due to the complexity of this task, performance is highly sensitive to random seeds, leading to generally lower success rates compared to previously reported results (Appendix F). In the sparse block stacking environment, the maximum number of blocks that can be stacked is five.
>
> [1] N., Taewook, et al. "Skill-based meta-reinforcement learning." ICLR, 2022.

---

### Decision · Program_Chairs · 2025-05-01

**Decision:**

Accept (poster)

**Comment:**

This submission received 3 accepts and 1 weak reject, with all reviewers recognizing that the paper introduces a well-motivated and practical approach to skill termination in RL through Novelty-Based Decision Point Identification (NBDI). Strengths highlighted include strong empirical performance across diverse environments, clear methodology, insightful visualizations, and the effective application of state-action novelty for variable-length skill learning. While initial concerns were raised regarding the breadth of evaluated baselines, sensitivity to dataset quality, hyperparameter tuning (e.g., novelty thresholds), the authors provided detailed rebuttals, additional analyses, and clarifications, for example adaptive thresholding strategies, broader applicability to other skill-based methods, and improved experimental explanations. Overall, most reviewers agreed that this paper offers meaningful contributions to skill-based RL, particularly in enhancing generalization and efficiency through learned termination conditions. The AC would have given a higher recommendation if the submission provided more thorough evaluations on more recent benchmarks compared to the ones in SPiRL.